# RARITY SCORE : A NEW METRIC TO EVALUATE THE UNCOMMONNESS OF SYNTHESIZED IMAGES

**Jiyeon Han**[1], **Hwanil Choi**[1], **Yunjey Choi**[2], **Junho Kim**[2], **Jung-Woo Ha**[2], **Jaesik Choi**[1]
[1] Kim Jaechul Graduate School of AI, KAIST
[2] NAVER AI Lab
{j.han, hwanil.choi, jaesik.choi}@kaist.ac.kr,
{yunjey.choi, jhkim.ai, jungwoo.ha}@navercorp.com

## ABSTRACT

Evaluation metrics in image synthesis play a key role to measure performances of generative models. However, most metrics mainly focus on image fidelity. Existing diversity metrics are derived by comparing distributions, and thus they cannot quantify the diversity or rarity degree of each generated image. In this work, we propose a new evaluation metric, called 'rarity score', to measure both image-wise uncommonness and model-wise diversified generation performance. We first show empirical observation that typical samples are close to each other and distinctive samples are far from each other in nearest-neighbor distances on latent spaces represented by feature extractor networks such as VGG16. We then show that one can effectively filter typical or distinctive samples with the proposed metric. We also use our metric to demonstrate that the extent to which different generative models produce rare images can be effectively compared. Further, our metric can be used to compare rarities between datasets that share the same concept such as CelebA-HQ and FFHQ. Finally, we analyze the use of metrics in different designs of feature extractors to better understand the relationship between feature spaces and resulting high-rarity images. Code will be publicly available for the research community.[1]

## 1 INTRODUCTION

Generative models have attracted a lot of attention for recent years. Generative Adversarial Networks (GANs) (Goodfellow et al., 2014) have achieved significant advancement over the past several years, enabling many computer vision tasks such as image manipulation (Bau et al., 2019; Jahanian et al., 2020; Shen et al., 2020; Härkönen et al., 2020; Kim et al., 2021), domain translation (Isola et al., 2017; Zhu et al., 2017; Choi et al., 2018; Kim et al., 2019; 2020; Choi et al., 2020), and image or video generation (Tulyakov et al., 2018; Karras et al., 2019; 2020b;a; 2021; Skorokhodov et al., 2022; Tian et al., 2021; Kim et al., 2022; Kim & Ha, 2022; Lee et al., 2022; Yu et al., 2022b). The emergence of diffusion models accelerates the advancements of the generative models especially in the field of text-to-image modeling (Ramesh et al., 2022; Saharia et al., 2022; Yu et al., 2022a; Rombach et al., 2022).

To quantify the performance of generative models, various metrics have been proposed. As standard evaluation metrics, inception score (IS) (Salimans et al., 2016), kernel inception distance (KID) (Bińkowski et al., 2018), and Frechét inception distance (FID) (Heusel et al., 2017) are prevalent for evaluating the quality of images synthesized by generative models. These metrics evaluate the discrepancy between generated and real image sets on the feature space characterized by a generative model with respect to diversity and fidelity. The fidelity represents the quality of the generated image, and the diversity indicates the variety among the generated images that the generator creates without mode collapse similar to the distribution of the training datasets. To improve the fidelity and diversity, it is required that the distribution of generated images is similar to the real image distribution (Kynkäänniemi et al., 2019; Naeem et al., 2020).

---

[1]Code is available at https://github.com/hichoe95/Rarity-Score.

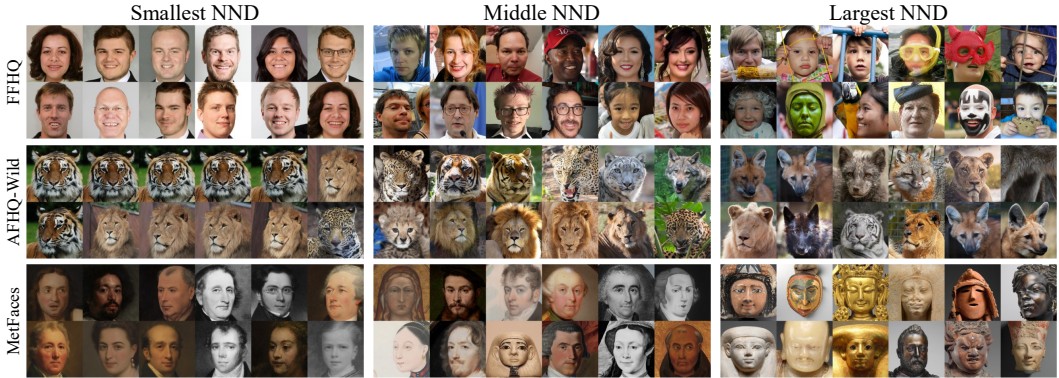

Figure 1: Real samples with the smallest nearest neighborhood distances (NNDs), middle NNDs, and the largest NNDs, respectively. For 'Middle NND' column, the images are randomly selected among the middle-ranked 200 images.

On the other hand, reproducing capability of the low density area is often overlooked. If we consider the distribution of training data, it is clear that the rare samples in the low density areas are poorly represented and thus likely to be difficult for the generator to reproduce. Examples of rare samples from public datasets include people with various accessories in FFHQ (Karras et al., 2019), white animals in AFHQ (Choi et al., 2020), and uncommon statues in MetFaces (Karras et al., 2020a).

Assessing the rarity is important not only because it is related to assessing the reproducing capability of the generative models, but also because it is related to selecting generated images. Recent commercialization of text-to-image models such as DALL-E 2 (Ramesh et al., 2022) and Stable Diffusion models (Rombach et al., 2022) enhance the necessity of metrics for uncommonness and creativity. In these creative AI application scenarios, a metric that measures each synthesized image (instance-wise metric) is essential for users and consumers to select images among the ones provided through API, rather than model-wise metrics such as FID (Heusel et al., 2017) and LPIPS (Zhang et al., 2018). Unfortunately, no instance-wise metric exists for measuring the creative and uncommon degree of each image despite its practicality and necessity.

In this paper, we propose a novel generative model evaluation metric that can represent the generative capabilities of rare samples in generative models as scores (a.k.a. rarity score). Our metric contributes to classifying rare images and typical images similar to those frequently observed in training datasets. The proposed rarity metric highlights the open problem of sparse generation of rare samples from generative models. Additionally, we have conducted comparative experiments on which of the previous state-of-the-art models produce more rare samples with preserving quality performances. Our contributions can be summarized as follows:

- We propose the first metric to quantify the rarity of individual generation which existing metrics cannot provide. Using the proposed metric, generations with the desired degree of rarity can be sampled.
- We show that the proposed metric can be used to compare the capability of generative models to generate rare samples. The proposed metric can further be used to compare which dataset contains more rare samples among the datasets that share the same concept such as CelebA-HQ and FFHQ datasets.
- We show the proposed metric can be applied on top of the various feature spaces with different viewpoints of rarity, by analyzing the sampled generations.

## 2 PRELIMINARIES

**Precision and Recall**   Precision and recall are commonly used performance metrics in many areas including classification tasks or natural language processing. In specific, to quantify the performance of the generative models, precision measures the fraction of the fake distribution that can

be generated from the real distribution. On the other hand, recall measures the fraction of the true distribution which can be reproduced by the fake distribution. Practically, $k$-nearest neighbor ($k$-NN) based method has been proposed to estimate the real and fake manifolds (Kynkäänniemi et al., 2019). For real samples $X_r \sim P_r$ and fake samples $X_g \sim P_g$, we first embed them in the high-dimensional feature space using pretrained DNNs such as VGG16 (andAndrew Zisserman, 2015) or the image encoder of CLIP (Radford et al., 2021) to get sets of feature vectors $\mathbf{\Phi_r}$ and $\mathbf{\Phi_g}$, respectively. The real and fake manifolds are estimated by the sets of $k$-NN spheres of each sample as follows.

$$\mathbf{manifold}_k(\mathbf{\Phi}) = \bigcup_{\phi_i \in \mathbf{\Phi}} B_k(\phi_i, \mathbf{\Phi}), \quad B_k(\phi_i, \mathbf{\Phi}) = \{\phi | d(\phi_i, \phi) \leq NN_k(\phi_i, \mathbf{\Phi})\} \tag{1}$$

Here, $NN_k(\phi_i, \mathbf{\Phi})$ represents the distance between $\phi_i$ and its $k$-th nearest neighbor in $\mathbf{\Phi}$. $B_k(\phi_i, \mathbf{\Phi})$ is the $k$-NN sphere of $\phi_i$ with the radius of $NN_k(\phi_i, \mathbf{\Phi})$ defined as a set of all $\phi$ whose distance to $\phi_i$ is smaller than or equal to $NN_k(\phi_i, \mathbf{\Phi})$. For the distance metric $d$, we use L2 distance for the rest of the paper. Then, precision and recall are respectively defined as

$$\text{precision}(\mathbf{\Phi_r}, \mathbf{\Phi_g}) = \frac{1}{|\mathbf{\Phi_g}|} \sum_{\phi_j \in \mathbf{\Phi_g}} \mathbb{I}(\phi_j \in \mathbf{manifold}_k(\mathbf{\Phi_r})), \tag{2}$$

$$\text{recall}(\mathbf{\Phi_r}, \mathbf{\Phi_g}) = \frac{1}{|\mathbf{\Phi_r}|} \sum_{\phi_i \in \mathbf{\Phi_r}} \mathbb{I}(\phi_i \in \mathbf{manifold}_k(\mathbf{\Phi_g})) \tag{3}$$

where $\mathbb{I}$ is indicator function. Precision often maps with the fidelity and recall maps to the diversity of the generations. However, it is nontrivial to apply both precision and recall to an individual generation as they are designed to measure on the sets of samples. To quantify the image quality of an individual generation, realism score has been proposed. Realism score measures the maximum of the inverse relative distance of a fake sample in a real $k$-NN sphere.

$$\text{realism score}(\phi_j) = \max_{\phi_i \in \mathbf{\Phi_r}} \frac{NN_k(\phi_i, \mathbf{\Phi_r})}{d(\phi_i, \phi_j)} \tag{4}$$

Realism score is high when the distance between the given fake sample and a real sample is relatively small compared to the radius of the $k$-NN sphere of the real sample. However, the realism score is not suitable to find rare samples as realism score can be large even if the radius of the $k$-NN sphere is small when the distance between fake sample and a real sample is also very small.

**Density and Coverage**   While precision and recall are commonly used, there are still several drawbacks that they are vulnerable to the outliers and computationally inefficient. To overcome the drawbacks, density and coverage have been proposed (Naeem et al., 2020). While precision only counts whether the fake sample is inside the real manifold or not, density counts the number of the real $k$-NN spheres that contains the fake sample.

$$\text{density}(\mathbf{\Phi_r}, \mathbf{\Phi_g}) = \frac{1}{k|\mathbf{\Phi_g}|} \sum_{\phi_j \in \mathbf{\Phi_g}} \sum_{\phi_i \in \mathbf{\Phi_r}} \mathbb{I}(\phi_j \in B_k(\phi_i, \mathbf{\Phi_r})) \tag{5}$$

If a fake sample is included in the multiple real $k$-NN spheres, it is more certain that the fake sample is inside the real manifold. Thus, density can be more robust to the outliers compared to the precision. On the other hand, similar to recall, coverage is relevant to the diversity of the generations. Coverage computes the number of real samples containing at least one fake sample in its $k$-NN sphere.

$$\text{coverage}(\mathbf{\Phi_r}, \mathbf{\Phi_g}) = \frac{1}{|\mathbf{\Phi_r}|} \sum_{\phi_i \in \mathbf{\Phi_r}} \mathbb{I}(\exists \phi_j \in \mathbf{\Phi_g}, \text{s.t.}, \phi_j \in B_k(\phi_i, \mathbf{\Phi_r})) \tag{6}$$

If a fake sample is sparsely located, the fake manifold can be exaggerated and the recall can be overestimated. As real manifold is known to have less outliers than the fake manifold, coverage can prevent such overestimation. Furthermore, coverage is more computationally efficient compared to the recall as coverage does not require the $k$-NN computations of fake samples. However, density and coverage are also not suitable to be applied on individual generations as they work on a set of generations.

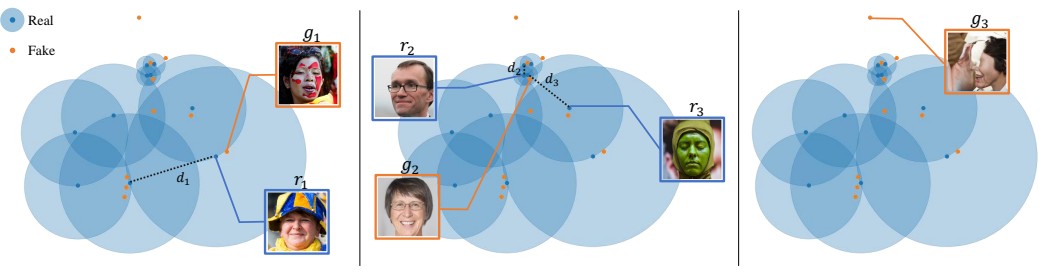

    (a) Case 1: Single sphere          (b) Case 2: Multiple spheres        (c) Case 3: Out of real manifold

Figure 2: Real and fake images and their embeddings in the feature space when calculating the rarity score. The transparent circle represents the $k$-NN sphere of the corresponding real sample at the center. (a) When $g_1$ is only in $k$-NN sphere of $r_1$, $g_1$ is scored by the radius of $r_1$ ($d_1$). (b) When $g_2$ is in both $k$-NN spheres of $r_2$ and $r_3$, $g_2$ is scored by the smallest radius ($d_2$). Since $g_1$ has the higher score than $g_2$, $g_1$ looks more unique than $g_2$. (c) If a sample is out of real manifold as $g_3$, we cannot guarantee the fidelity of the image and thus exclude it from the scoring procedure.

**Truncation Trick** The truncation trick is widely used in generative models to restrict latent sampling space to maintain the image quality (Kingma & Dhariwal, 2018; Brock et al., 2019; Karras et al., 2019). Typically, in StyleGAN based models (Karras et al., 2019; 2020b;a), the truncation trick is applied in the warped latent space $\mathcal{W}$ from the input latent space $\mathcal{Z}$ with $f : \mathcal{Z} \rightarrow \mathcal{W}$ as follows.

$$\mathbf{w}' = \bar{\mathbf{w}} + \psi(\mathbf{w} - \bar{\mathbf{w}}), \quad \bar{\mathbf{w}} = \mathbb{E}_{z \in P(z)}[f(z)] \tag{7}$$

When $\psi = 1$, it is the same as not using the truncation trick. As $\psi$ decreases, the latent code shifts toward the mean latent code. When $\psi = 0$, $\mathbf{w}'$ becomes $\bar{\mathbf{w}}$, which is the mean latent code in $\mathcal{W}$. Truncation trick is known to increase the fidelity at the expense of lowering the diversity of the generations.

## 3   RARITY SCORE

To evaluate the rarity of both individual generated instance and generative model performance, we propose a novel metric called *rarity score*. Following the idea of Kynkäänniemi et al. (2019) and Naeem et al. (2020), we use $k$-NN to represent the manifolds of real and generated samples as in Figure 2.

### 3.1   INSTANCE-WISE RARITY SCORE

We hypothesize that ordinary samples would be closer to each other whereas unique and rare samples would be sparsely located in the feature space. Figure 1 shows the real samples with both the smallest nearest neighbor distances (NNDs) and the largest NNDs from the training dataset. For all datasets, samples with the smallest NNDs show representative and typical images. On the contrary, the samples with the largest NNDs have strong individuality and are significantly different from the typical images with the smallest NNDs. From this intuition, we propose the rarity score:

$$Rarity(\phi_g, \mathbf{\Phi_r}) = \min_{r, s.t. \phi_g \in B_k(\phi_r, \mathbf{\Phi_r})} NN_k(\phi_r, \mathbf{\Phi_r}). \tag{8}$$

Equation (8) measures the rarity score of the given fake sample as the minimum radius of $k$-NN sphere of the real sample that contains the given fake sample. Equation (8) can be viewed as a dual-way k-NN, if we see the constraint $\phi_g \in B_k(\phi_r, \mathbf{\Phi_r})$ as a reverse k-NN (RkNN) (Korn & Muthukrishnan, 2000; Tao et al., 2004). RkNN is known as a problem to find the set of samples to which the given query is at least the k-th most influential. Thus we are filtering real samples to those that can be influential to the given fake sample. Equation (8) then takes 1-NN to the RkNN set.

For Case 1 in Figure 2, $g_1$ is in the $k$-NN sphere of $r_1$, thus the rarity score of $g_1$ is the radius of the $k$-NN sphere of $r_1$, which is $d_1$. Case 2 in Figure 2 shows that even though $g_2$ is in intersection

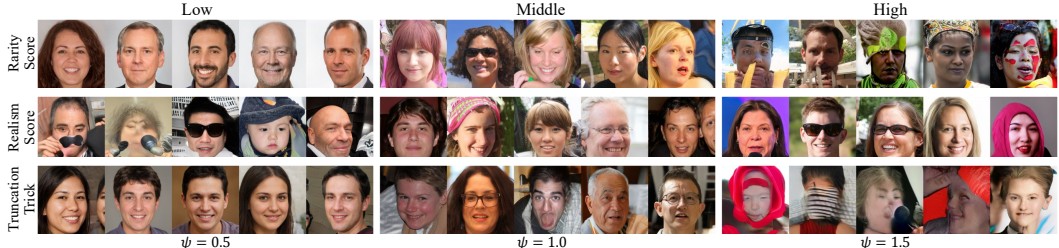

Figure 3: Images generated by StyleGAN2-FFHQ, sorted by the various methods. For the rarity score and the realism score, 'Low' column presents the generations with the top 5 lowest scores, 'High' column presents the generations with the top 5 highest scores, and 'Middle' column presents the random generations in the middle of the top 5th and the bottom 5th. For the truncation trick, 'Low', 'Middle', 'High' column show random generations with the truncation parameters $\psi = 0.5, 1.0, 1.5$, respectively.

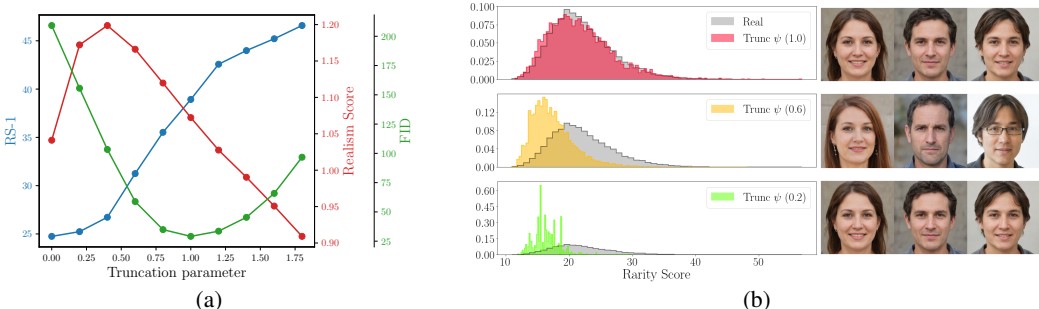

Figure 4: **(a)** RS-1 scores (blue), Realism scores (red), and FIDs (green) according to the truncation parameter $\psi$ for StyleGAN2-FFHQ. **(b)** Histograms of rarity score according to the truncation parameter $\psi$. The images next to the histogram are the generations for the corresponding $\psi$.

of k-NN spheres of $r_2$ and $r_3$, $g_2$ is scored by radius of $r_2$ ($d_2$). Since, in addition, $g_1$ looks more distinctive compared to $g_2$, it has higher rarity score. As samples outside of the real manifold cannot guarantee their high fidelity, our metric does not consider the generations not contained in the real manifold **manifold**$_k(\mathbf{\Phi_r})$. For example, the generation $g_3$ in Figure 2 (c) is an artifact and it is out of the real manifold. Thus the rarity score for $g_3$ is undefined. This ensures the fidelity of the selected images without any additional fidelity metrics. The synthesized examples out of the real manifold are presented in Appendix E. Throughout this paper, we use $k = 3$ for the experiments, which is empirically chosen with the supports in Appendix G.

## 3.2 MODEL-WISE RARITY SCORE

**Model-wise Measurement** As rarity score is defined as an instance-wise measurement, there can be many ways to aggregate the rarity scores of a set of fake samples to compare between generative models. In this paper, we suggest one way of model-wise measurement, *RS-p*, based on the rarity score.

$$\text{RS-p}(\Theta) := \frac{1}{|\Phi_p|} \sum_{\phi_g \in \Phi_p} Rarity(\phi_g), \quad \Phi_p = \{\phi_g \in \Phi_g^\Theta | CDF(Rarity(\phi_g)) \geq 1 - p/100\}, \quad (9)$$

where $\Phi_g^\Theta$ denotes the set of fake samples generated by a model $\Theta$. We omit $\Phi_r$ in *Rarity* as it is fixed. Basically, RS-p is the mean rarity score of the fake samples with the top p% rarity scores. RS-p is designed to compare the rarity of the most rare generations.

**Relationship with Existing Metrics** Here, we mainly investigate the relationship between our rarity and two existing methods that are most relevant to ours: the realism score (Kynkäänniemi et al., 2019) and the truncation trick (Brock et al., 2019). The realism score is a metric which measures the fidelity of an individual generation. While the realism score takes into account the

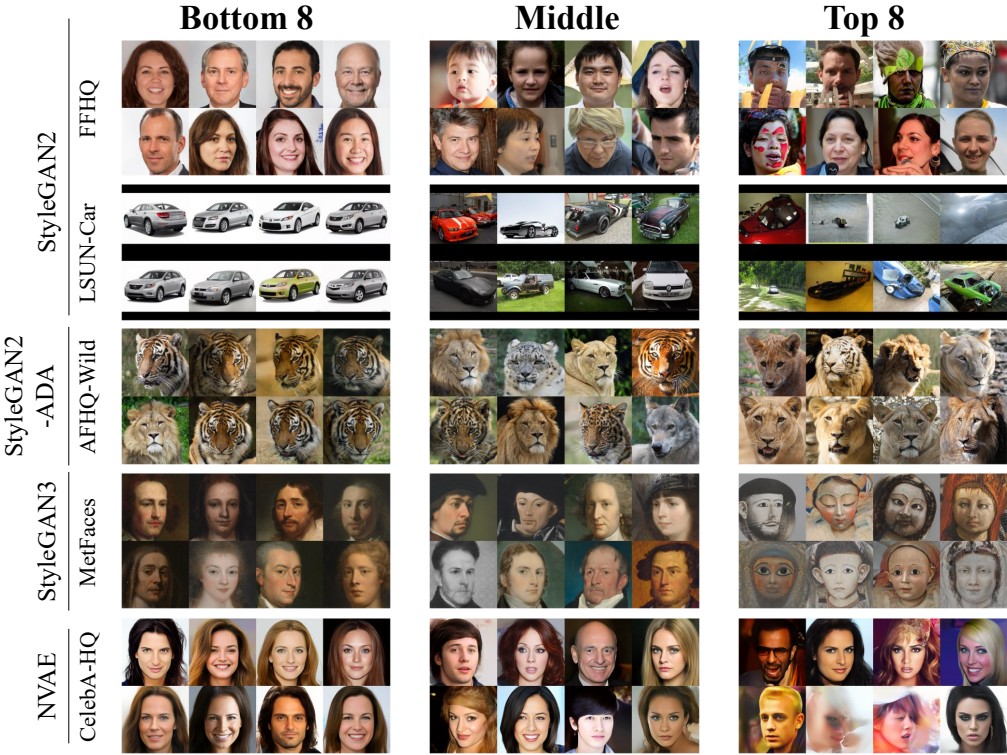

Figure 5: Bottom 8, Top 8, and Middle 8 generated images in terms of the proposed rarity score on five datasets and four generation models.

radius of the real $k$-NN sphere, the effect of the radius is degraded by the effect of the distance between the real and the fake sample. Realism score can be high for the samples in the high density region which is assumed to be typical and thus not suitable to measure rarity. In Figure 3, while images with the high realism scores show high fidelity, the images with the high rarity scores show more extraordinary characteristics in the images. We further compare our metric with the inverse realism score by conducting a user study. The results are summarized in Appendix I.3.

Truncation trick has been proposed to maintain high fidelity of generations by filtering out the outskirt latent codes far from the mean latent code. This method can increase the fidelity of the images, but it can compromise its versatility. If we use $\psi \geq 1$ to shift the latent code in the opposite direction from the mean latent code, we might get more diverse images. In this case, however, we cannot guarantee the fidelity of images as shown in Figure 3.

As it is known that the truncation trick shows diversity-fidelity trade-off, it will be convenient to compare with other metrics based on the truncation parameter changes. When truncation parameter increases, one can presume that the diversity will be increased and likely the rarity of generations as in Figure 4. This is well shown in Figure 4a, where rarity score increases as the truncation parameter increases. However, it is hard to be captured with FID or realism score, as (1) FID penalizes when the distribution is far from that of real as in the cases of $\psi > 1.0$ and (2) realism score is more focused on the fidelity. On the other hand, as rarity score does not measure the fidelity directly, it will be helpful to use rarity score along with the fidelity metric, for selecting images or models.

## 4 EXPERIMENTAL RESULTS

In this section, we present the experimental results of the proposed metric for various standard GANs and datasets. We use $k = 3$ for the manifold approximation throughout the experiments. We use 30k real images to approximate the real manifold and calculate the rarity of 10k fake images. We use VGG16 as a feature extractor for all experiments except for Section 4.4.

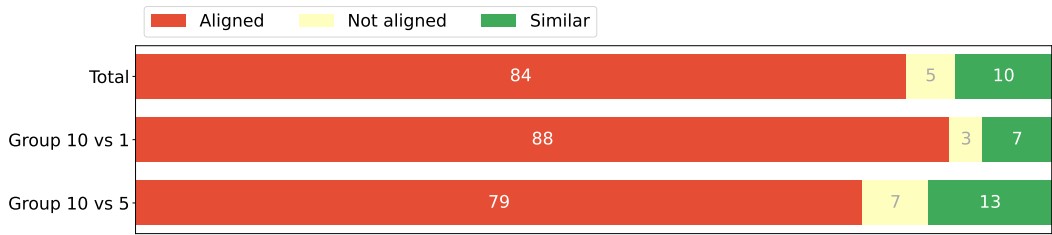

Figure 6: Alignment of human evaluation and rarity score.

## 4.1 RARITY OF INDIVIDUAL GENERATION

Figure 5 shows synthesized images with top 8 highest, top 8 lowest, and 8 middle rarity scores. As expected, the generated images with the lowest rarity scores are the ones that are typical in the training dataset. On the other hand, the generations with the highest rarity scores are far from typical images and have various characteristics and uniqueness that are rare in the training dataset. For FFHQ dataset, for example, we can see clear and formal human face images in the 'Bottom 8' column whereas we can see elaborated face images such as face with a colorful makeup or a fancy hat in the 'Top 8' column. For AFHQ-Wild dataset, bottom images mostly show the faces of typical tigers while top images are mostly female lions with light-colored fur, which are far from the typical lion. Further, MetFaces dataset shows oil portraits in bottom images while top images show statues or flat paintings. However, for the datasets which are uncurated and not aligned such as LSUN datasets, the top 8 images are not clear as that of the aligned datasets because the training dataset contains a lot of noisy images.

**A User Study on Rarity Score** We have conducted a user study on whether our proposed rarity score aligns with the human perception of uncommonness. 61 participants answered our survey. In this survey, we target on StyleGAN2-FFHQ model. The results of user study are summarized in Figure 6 and the details of the experimental setting is described in Appendix I.

In total, 84% of answers were aligned with the rarity score, i.e., people interpret images with the lower rarity scores as more common images. For more challenging comparison between Group 10 (the highest rarity group) and Group 5 (middle rarity group), the alignment is lesser than that of Group 10 and Group 1 (the lowest rarity group) and the portion of answer "Both sets are similar" increases. This suggests that the proposed metric can distinguish the degree of difference in commonness.

## 4.2 RARITY OF GENERATIVE MODELS

Not only the proposed metric can be applied on an individual generation, it can be used to compare generative models on the same dataset which shares the real manifold. While there exist diversity metrics on the generative models such as LPIPS (Zhang et al., 2018), our method is different in that it focuses on the ability to generate unique and rare generations while the existing metrics focus on diversity between generations.

**An Empirical Comparison between Generative Models** We further compare the generative models using the rarity score. Since our main focus is on the rare samples, we compare models by the RS-p score which is the mean rarity score of the top $p\%$ samples among 10000 samples. The left graph in Figure 7 shows the RS-p scores of the various models as $p$ changes from 0.1 to 1.0. The images in the right column of Figure 7 are the top 10 images with the highest scores, which corresponds to $p = 0.1$. Under $p = 0.1$, StyleGAN2 shows the highest RS-p score even when compared with the real images as we can see in the generations on the right. The images generated by StyleGAN2 are colorful and have diverse accessories, such as crown, mic, and hair band. On the other hand, the images from UnetGAN seem to have relatively less diversity. We can interpret that UnetGAN generates images more conservatively as its RS-p scores are the lowest among the models for all $0.1 \leq p \leq 1.0$. We find that the OOM ratio does not affect much to the RS-p score when

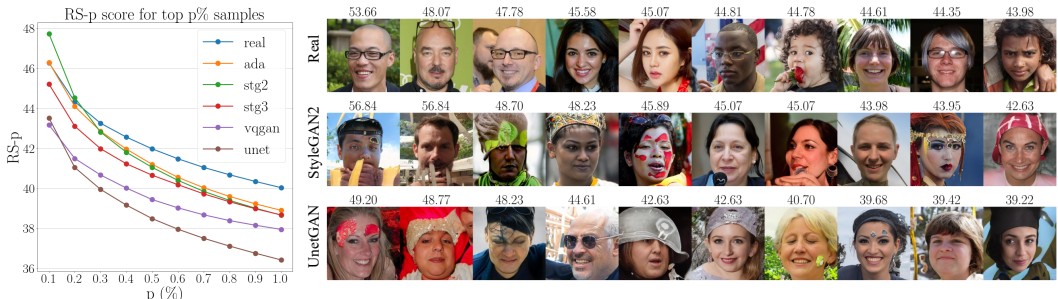

Figure 7: RS-p score of the top $p\%$ samples (left) and the top 10 images with the highest rarity scores (right). The number above each image represents the rarity score of the corresponding image. We investigate the mean scores of top $p\%$ samples for 5 models trained on FFHQ dataset; StyleGAN (stg), StyleGAN2 (stg2), StyleGAN3 (stg3), VQGAN (vqgan), and UnetGAN (unet).

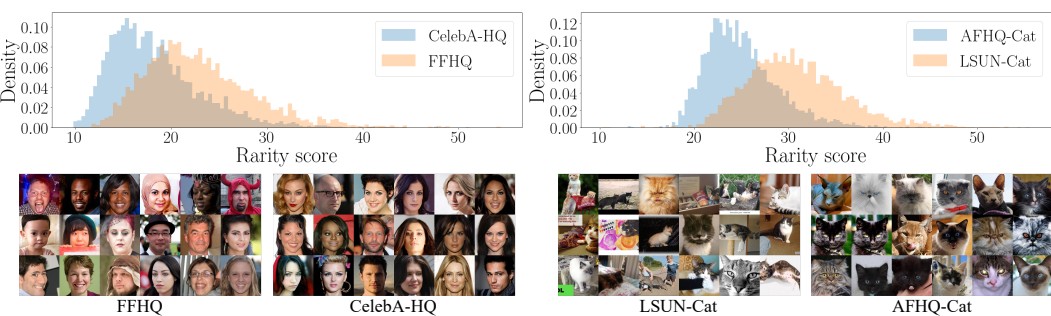

Figure 8: The density of rarity score for 10000 images sampled from datasets sharing the same concept (top) and images with the high rarity scores for each dataset (bottom). Rarity score is measured on the union of real manifolds for a pair of datasets.

VGG16 is the feature extractor. We further observe that high RS-p score does not always imply low fidelity, which is further discussed in Appendix N.

## 4.3 COMPARISONS BETWEEN SIMILAR DATASETS

We also propose a method to compare rarity between two different datasets which share the same concept, such as CelebA-HQ and FFHQ for human faces. Rarity scores of each dataset are calculated on the union of two real datasets, $\mathbf{\Phi_r} = \mathbf{\Phi_{r_1}} \cup \mathbf{\Phi_{r_2}}$. For example, when we calculate the scores of human face datasets such as CelebA-HQ and FFHQ, the real manifold $\mathbf{manifold}_k(\mathbf{\Phi_r})$ is $\mathbf{manifold}_k(\mathbf{\Phi_{CelebA-HQ}} \cup \mathbf{\Phi_{FFHQ}})$. Figure 8 shows comparisons on normalized rarity score for two sets of datasets that have the same concept. In the case of human face dataset, the density of CelebA-HQ is skewed left while the density of FFHQ is widely distributed to the right. It can be interpreted as FFHQ has more uncommon images than CelebA-HQ. In other words, there are less uncommon images in CelebA-HQ. This makes sense because CelebA-HQ is collected among the celebrities while FFHQ is collected among the public with considerable variation in terms of age, ethnicity, and image background. The images in Figure 8 are top scoring images in each dataset. Since the maximum rarity score of FFHQ is greater than that of CelebA-HQ, the top images of FFHQ are relatively more uncommon. Similarly, LSUN-Cat and AFHQ-Cat are both datasets on cat images where LSUN-Cat is uncurated cat images and AFHQ-Cat is curated with close-up cat faces only. In Figure 8, we can confirm that LSUN-Cat is more diverse than AFHQ-Cat as the rarity score distribution of AFHQ-Cat is more on the left and concentrated compared to the distribution of AFHQ-Cat.

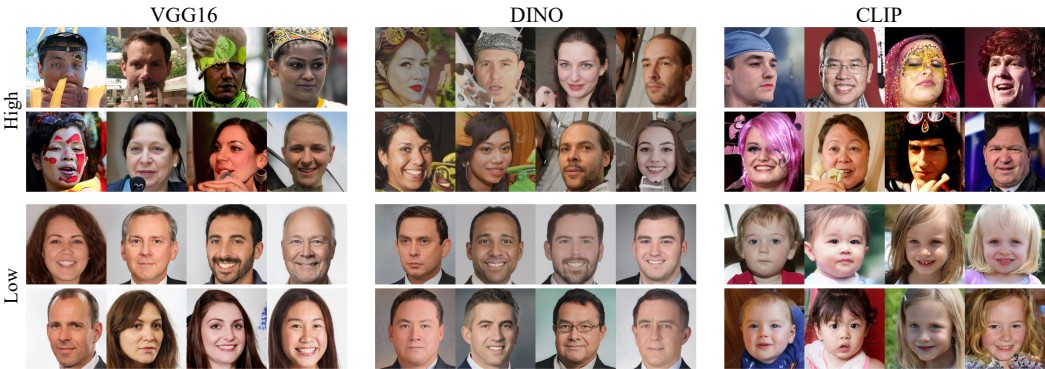

Figure 9: StyleGAN2 generations of FFHQ with the highest rarity scores and the lowest rarity scores for various feature extractors. For the low score images, however, as CLIP is one of the self-supervised models, it captures semantics rather than structures or patterns unlike the other classifiers.

## 4.4 A STUDY ON FEATURE EXTRACTORS

Since the proposed metric operates in the feature space, the perspective of the rarity may vary depending on which feature extractor is used. Figure 9 presents images with high and low rarity scores for diverse feature extractors to analyze the effects of the extractors. As a classifier trained on the Imagenet dataset, VGG16 focuses more on the objects and the patterns. There are many images with colorful face painting among the high score samples with VGG16. On the other hand, ViT models such as DINO (Caron et al., 2021) are known to focus more on the structure compared to the CNN-based models (Park & Kim, 2022). This can be seen from the second column of Figure 9. The DINO-based high score images have various face angles while the low score images are mostly front-facing. In the third column of Figure 9, the baby faces appear in the low score images, unlike the other two models which have formal adult images in the low score. This can be explained from the previous study that the CLIP model (Radford et al., 2021), which is a self-supervised image-text multi-modal model, can better capture semantics compared to the classifiers. We conjecture that the model may consider the baby faces more similar between them than between the other images because baby faces are less distinctive and share a lot common properties compared to the adult faces. For the user preference, we present a user study on the feature extractors for FFHQ dataset and found that VGG16 is the most preferred feature extractor for FFHQ dataset. The detailed experimental setting and results are described in Appendix I. For the noise robustness, on the other hand, we found CLIP is more robust to the Gaussian noise or Gaussian blur compared to VGG16 for various noise levels. The details are described in Appendix J.

## 5 CONCLUSION

In this paper, we propose a new evaluation metric called *rarity score* to measure how rare a synthesized image is based on the real data distribution. Each generated image is scored by estimating the density around the target image on the real manifold. Our rarity score enables several following tasks that existing metrics, such as FID and Precision&Recall, cannot do. First, the proposed metric can evaluate the rarity or diversity of an individual image rather than a set of images. Second, the proposed metric can be applied to compare between generative models or datasets that share the same concept in terms of the density of the rare samples. In addition, we provide studies on feature extractors that the viewpoint of commonness for each feature extractor varies depending on its structure (e.g. fully CNN, ViT) or how it was learned (e.g. fully supervised, self-supervised). The main limitation of our method is that it relies on the training dataset. For example, for the LSUN datasets which have non-aligned images, noise images can have high rarity scores, which would not be what users expect to have strong characteristics for the target category. Further, it might reflect the bias that the training dataset may contain. However, this can be useful for studying fairness and bias in the datasets or the generative models.

## ETHICS STATEMENT

It is challenging to quantify the creativity and is not exceptional for AI's creativity. Rarity score that we propose can be regarded as a step toward quantifying creativity even if it focuses on uncommonness and uniqueness based on distribution distance. To achieve better creativity quantification, it is essential to make a consensus among more multidisciplinary experts including artists, philosophers, social scientists as well as AI researchers.

Since our study is to propose a metric for evaluating image synthesis and manipulation models, it is inevitable to perform user study to validate the efficacy and robustness of the proposed Rarity Score. Considering that our user study is to simply compare the diversity of images generated by each model and rarely give participants stress, we did not apply IRB review process. However, we acknowledge Ethical review board and agree their comments that this decision needs to be approved by IRB and it is necessary to make a more concrete guideline of IRB process in machine learning and data science research community.

## REPRODUCIBILITY STATEMENT

We will release our code and the implementation details to be publicly available for reproducibility. We will also include the url links to access various feature extractors.

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

# A    RANK CORRELATIONS

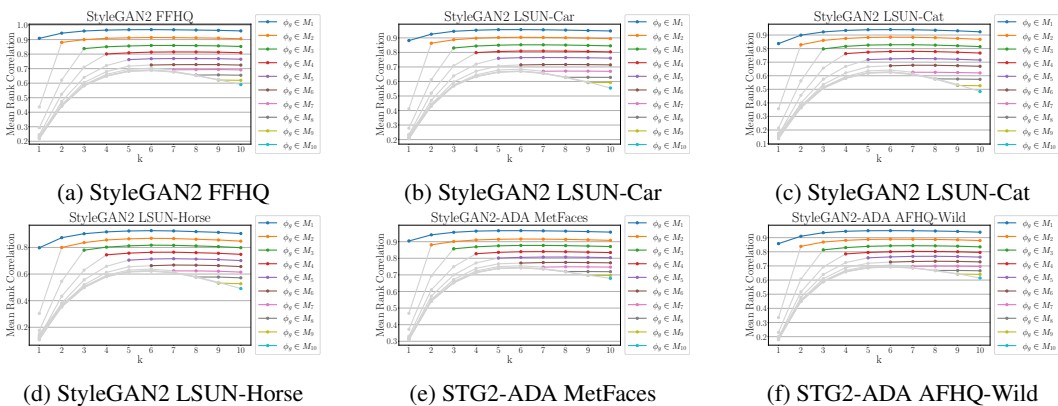

(a) StyleGAN2 FFHQ     (b) StyleGAN2 LSUN-Car     (c) StyleGAN2 LSUN-Cat

(d) StyleGAN2 LSUN-Horse     (e) STG2-ADA MetFaces     (f) STG2-ADA AFHQ-Wild

Figure 10: Rank correlations for the various models. STG2 stands for StyleGAN2.

# B    DISTRIBUTIONS ACCORDING TO THE TRUNCATION PARAMETER

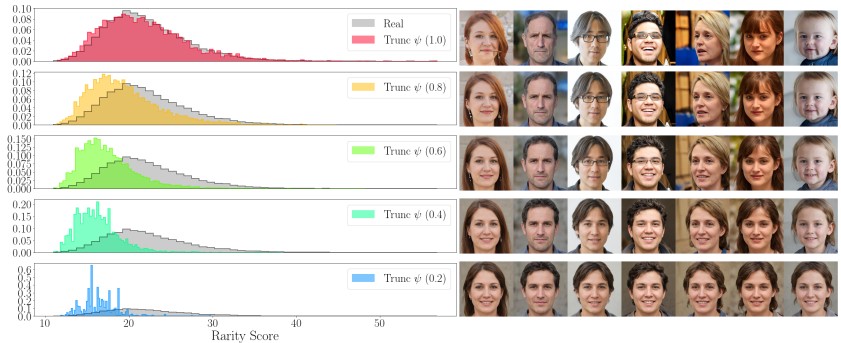

Figure 11: Histograms of the rarity score for 10000 generations varying the trunctaion parameter $\psi$ from 1.0 to 0.2. As expected, the distribution moves to the left as $\psi$ gets smaller (less diverse).

# C    COMPARISONS BETWEEN MODELS

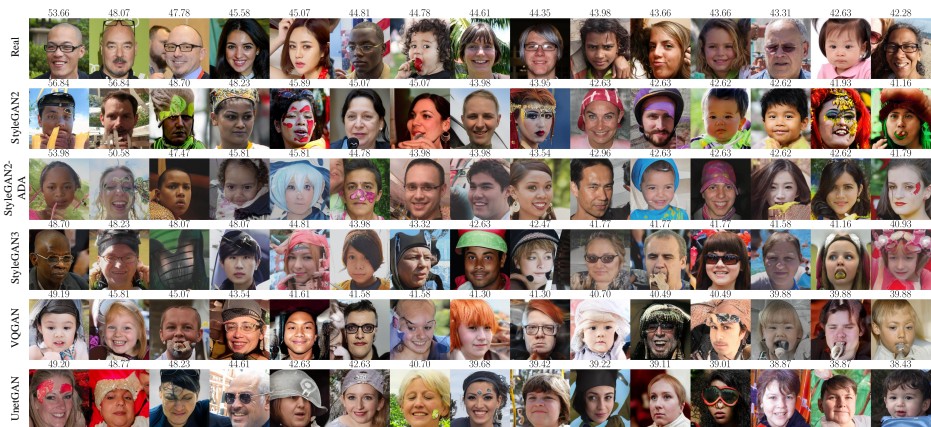

Figure 12: Top 15 samples of the highest rarity scores for the models trained on FFHQ dataset.

## D REAL SAMPLES ORDERED BY NND AND FAKE SAMPLES ORDERED BY RARITY SCORE

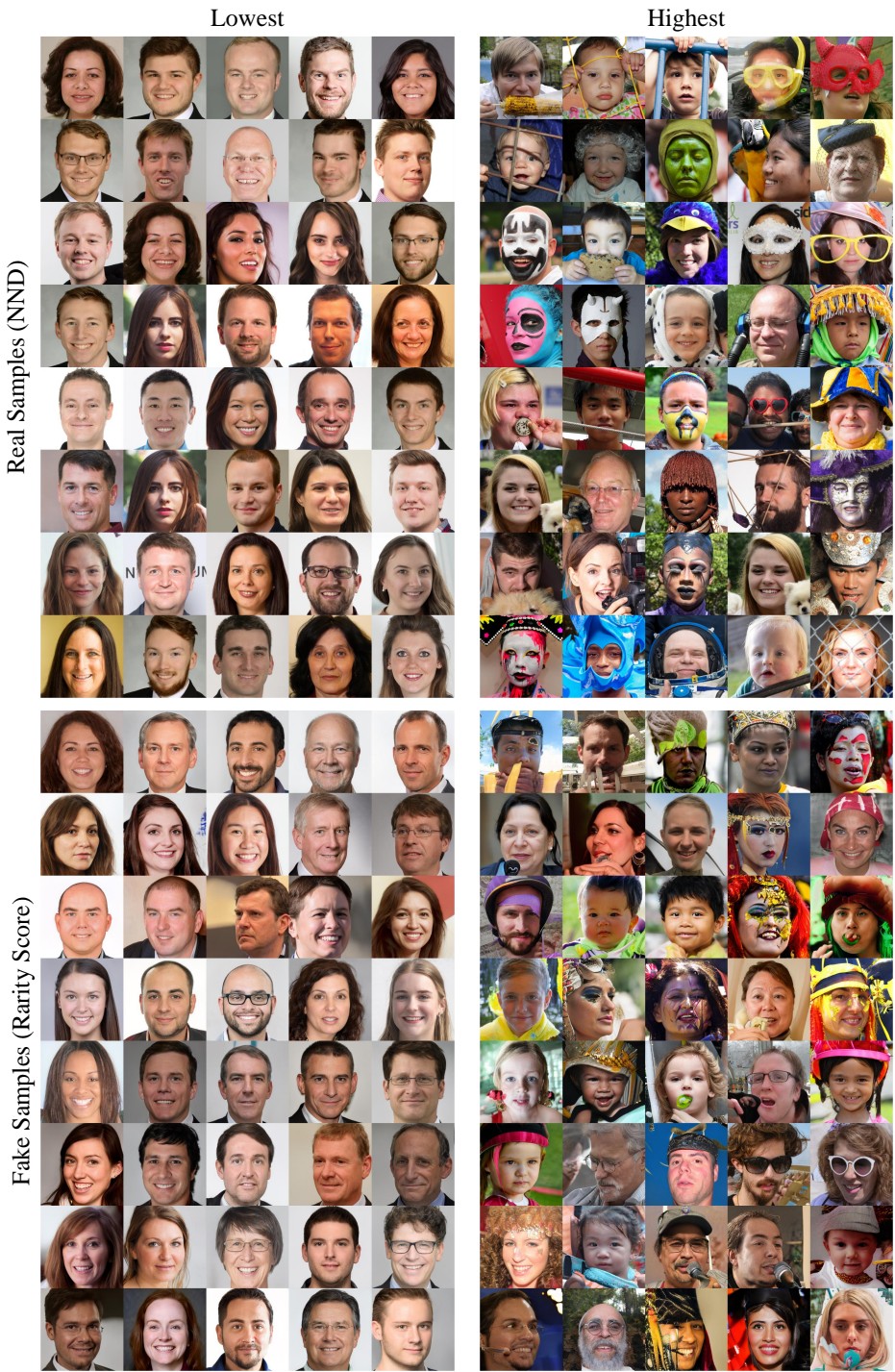

Figure 13: Real samples with the lowest/highest NND from FFHQ dataset (top) and fake samples generated from StyleGAN2 with the lowest/highest rarity score (bottom).

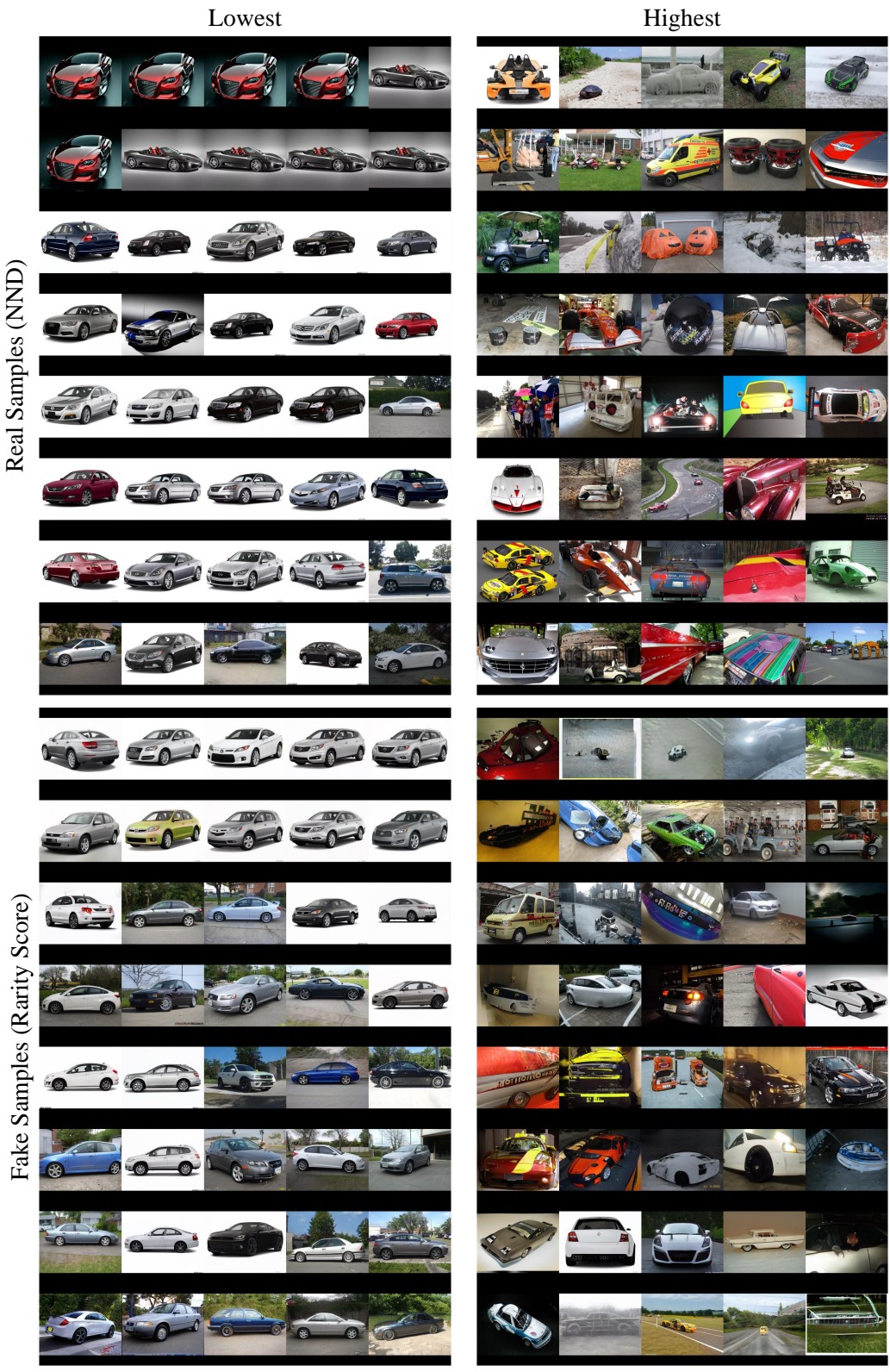

Figure 14: Real samples with the lowest/highest NND from LSUN-Car dataset (top) and fake samples generated from StyleGAN2 with the lowest/highest rarity score (bottom).

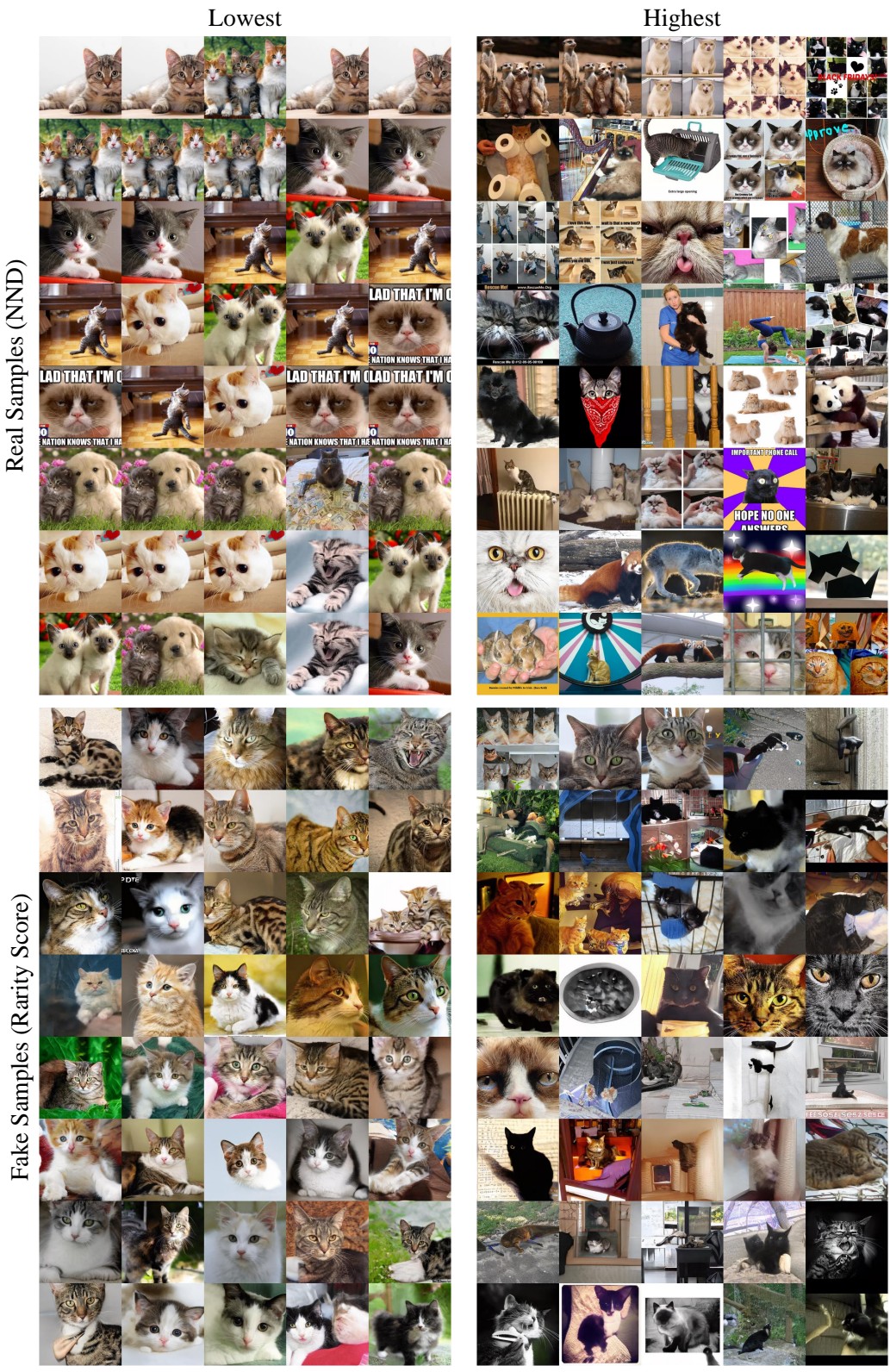

Figure 15: Real samples with the lowest/highest NND from LSUN-Cat dataset (top) and fake samples generated from StyleGAN2 with the lowest/highest rarity score (bottom).

Lowest        Highest

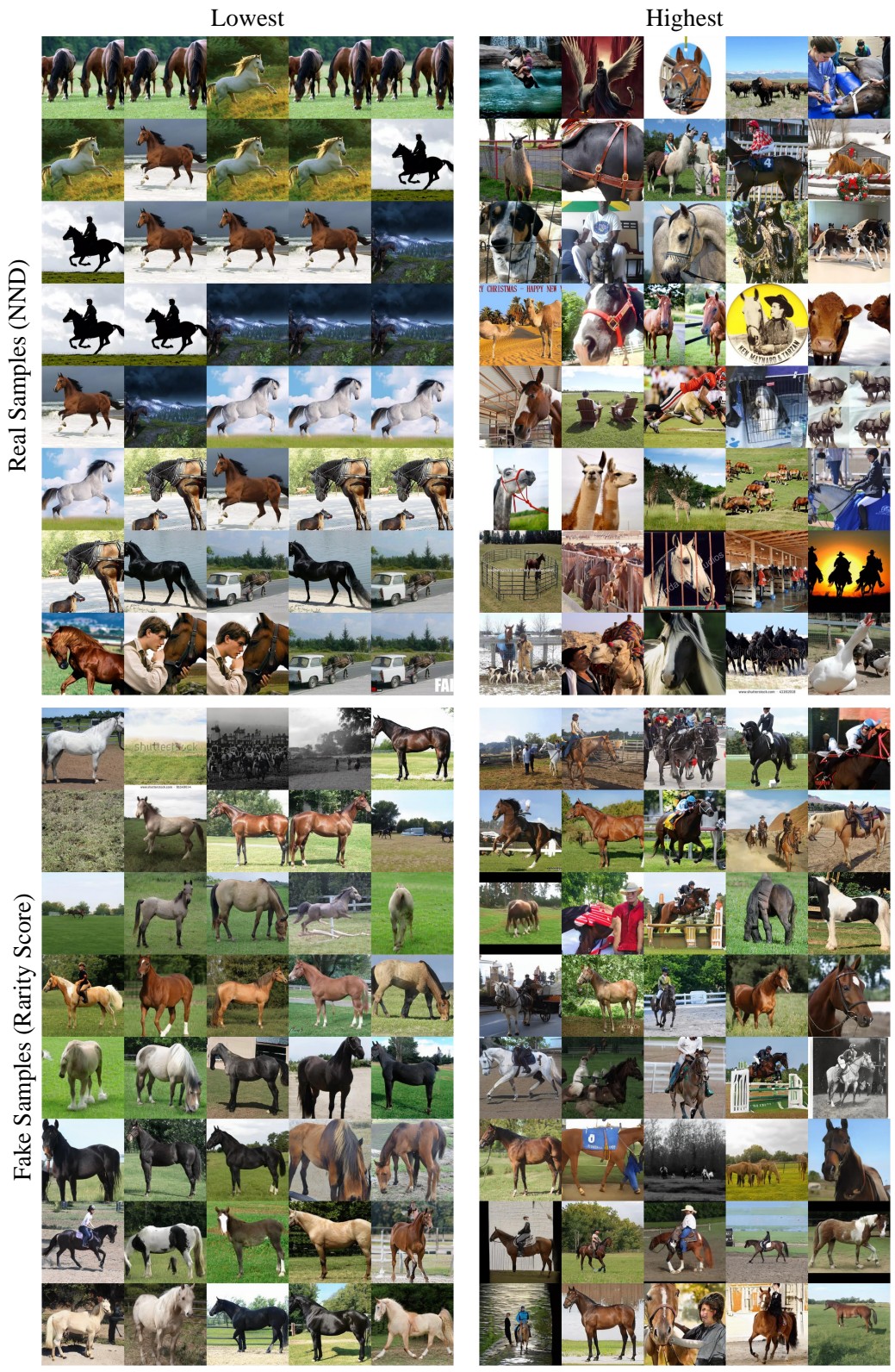

Figure 16: Real samples with the lowest/highest NND from LSUN-Horse dataset (top) and fake samples generated from StyleGAN2 with the lowest/highest rarity score (bottom).

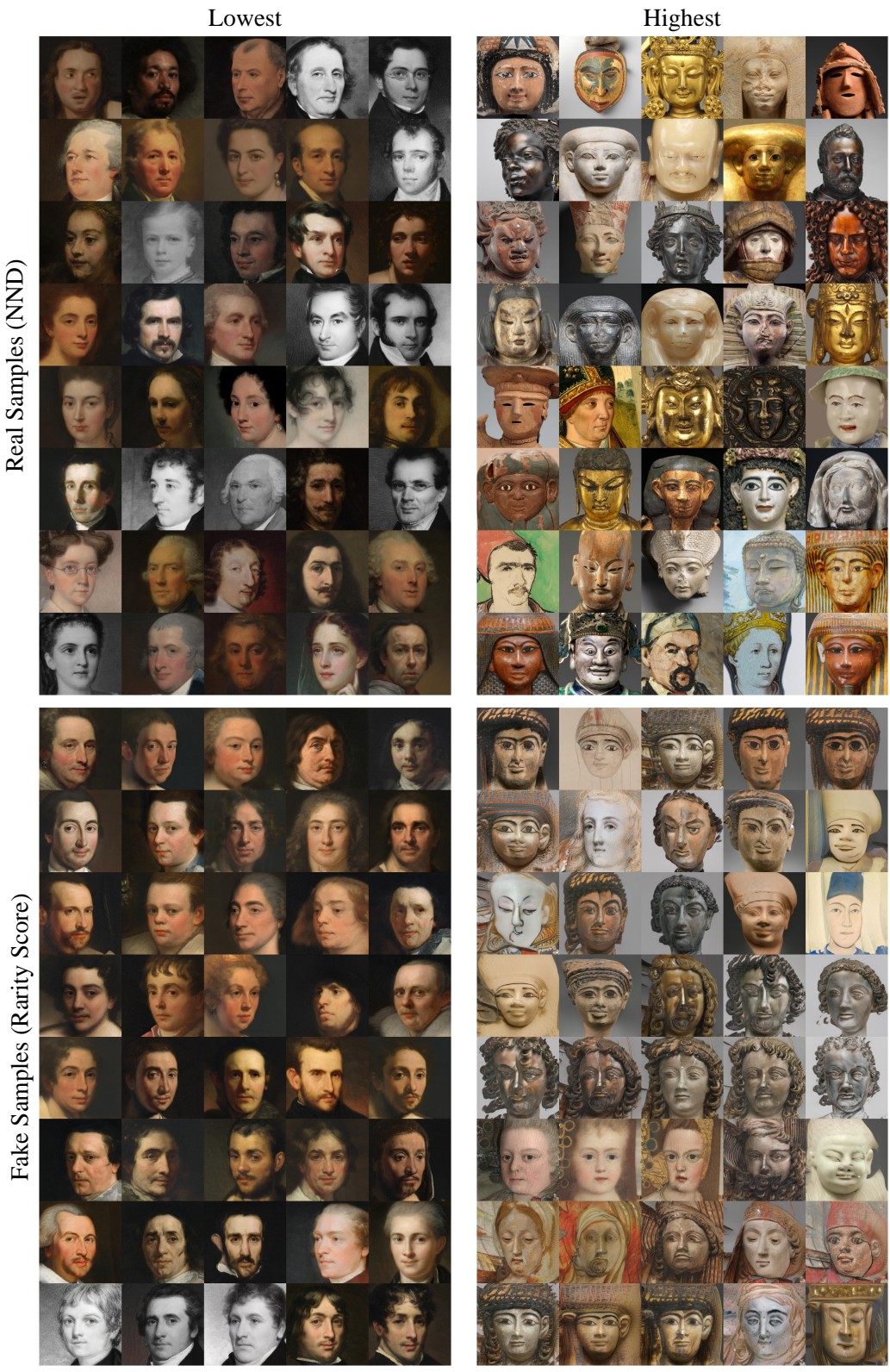

Figure 17: Real samples with the lowest/highest NND from MetFaces dataset (top) and fake samples generated from StyleGAN2-ADA with the lowest/highest rarity score (bottom).

Lowest                                    Highest

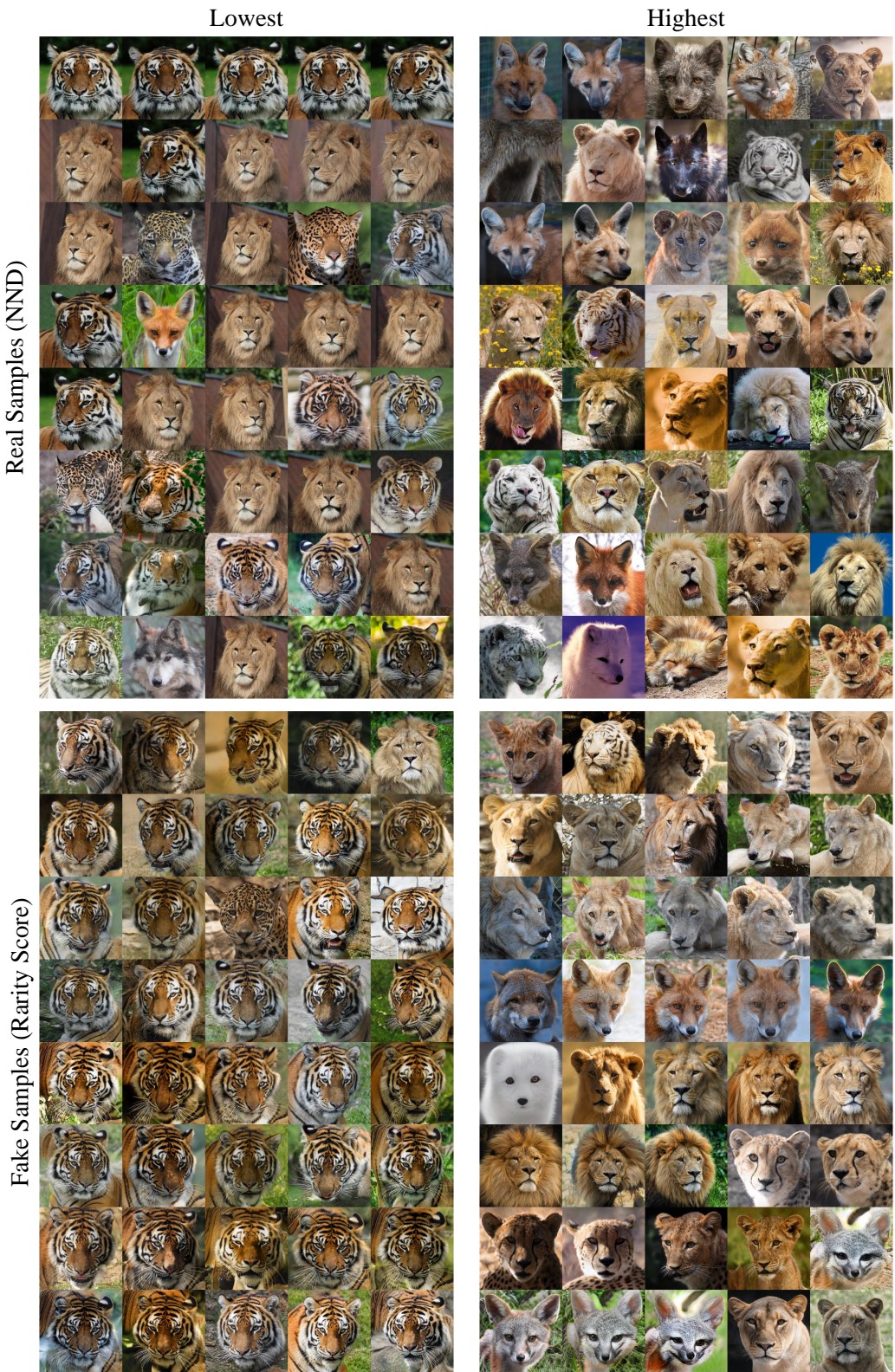

Figure 18: Real samples with the lowest/highest NND from AFHQ-Wild dataset (top) and fake samples generated from StyleGAN2-ADA with the lowest/highest rarity score (bottom).

# E    EXAMPLES OF OUT-OF-MANIFOLD SAMPLES

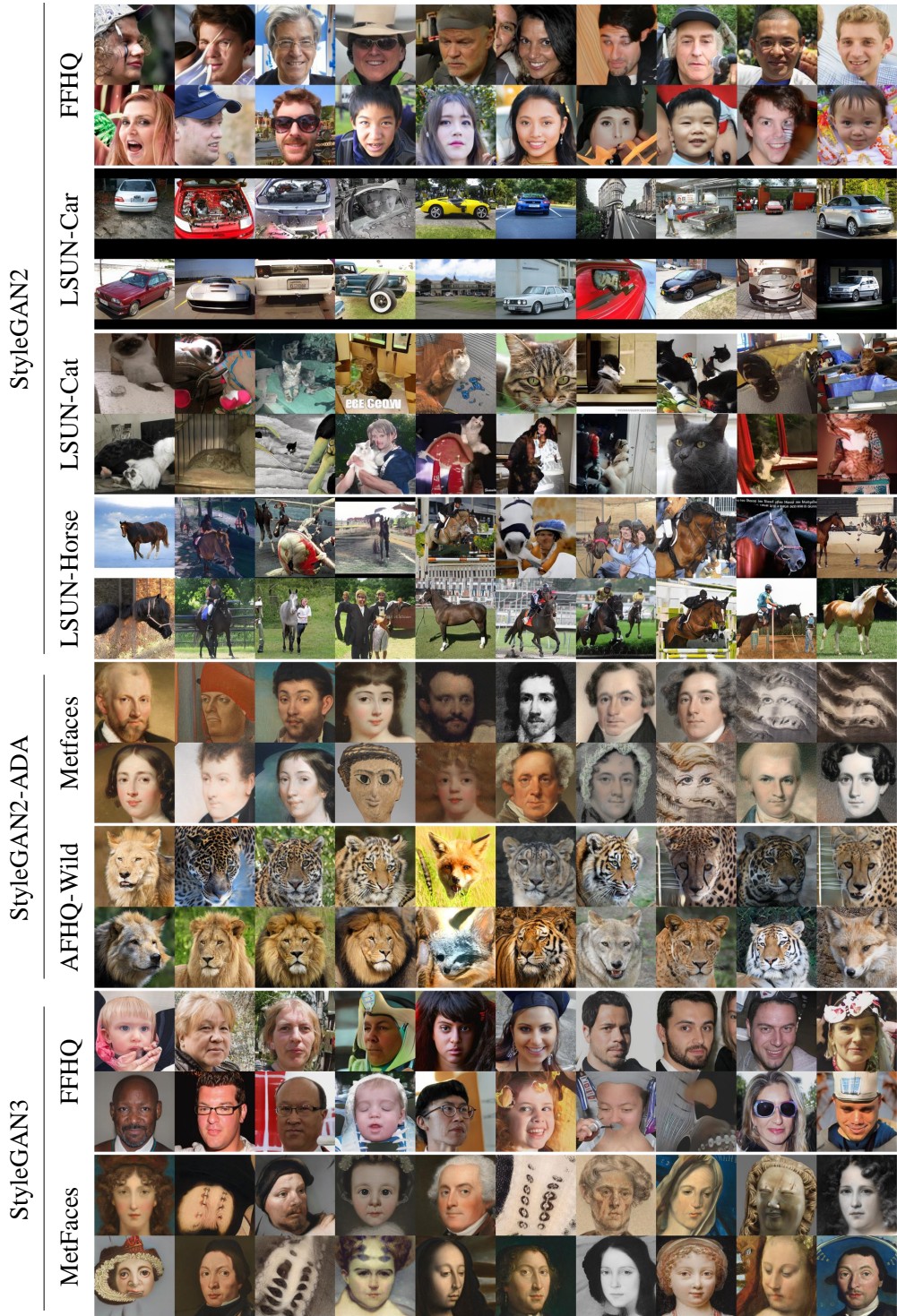

Figure 19: Out-of-manifold samples for various models.

# F RESULTS FOR THE VARIOUS FEATURE EXTRACTORS

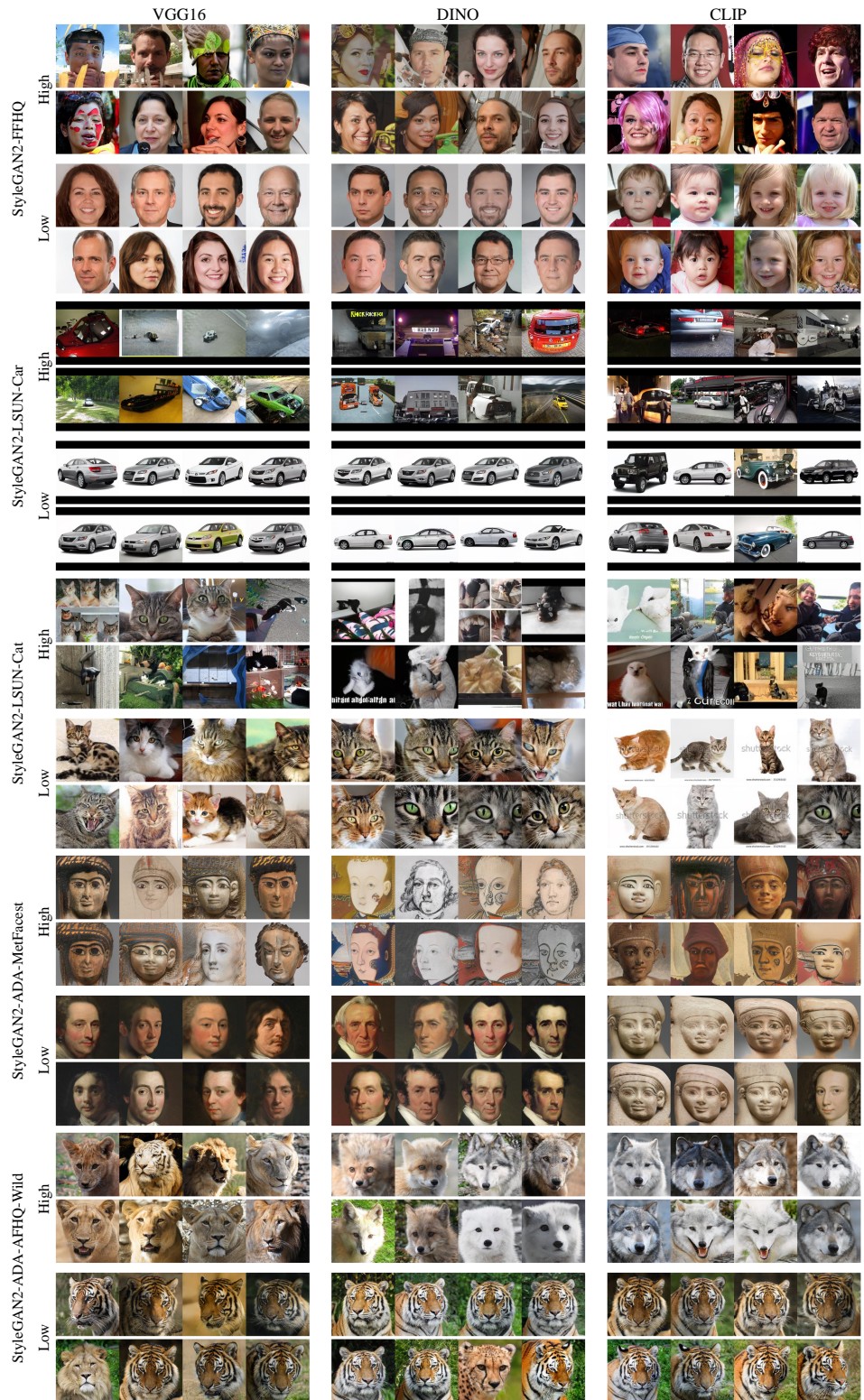

Figure 20: Fake samples with the lowest/highest rarity scores based on various feature extractors.

# G  SELECTION OF $k$

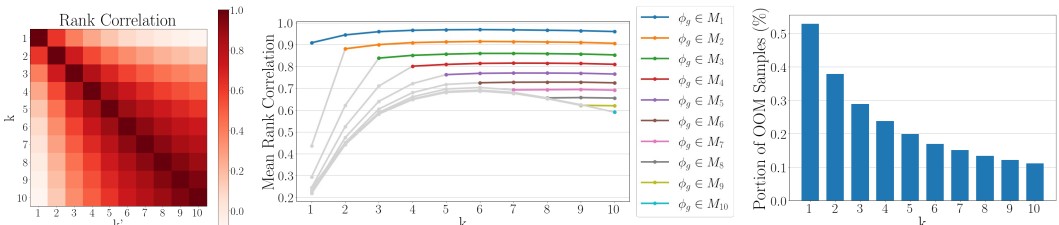

Figure 21: Rank correlations between $1 \leq k \leq 10$ for the rarity scores. Left : Rank correlation matrix between $1 \leq k \leq 10$ and $1 \leq k' \leq 10$ for the fake samples in $\{\phi_g|\phi_g \in M_1\}$ where $M_i$ stands for **manifold**$_i(\mathbf{\Phi_r})$) with $i$ as the $k$-NN parameter. Center : Mean rank correlation for all $k$s. Each plot represents the mean rank correlation for the fake samples in $\{\phi_g|\phi_g \in M_i\}$. Right : Portion of out of manifold (OOM) samples.

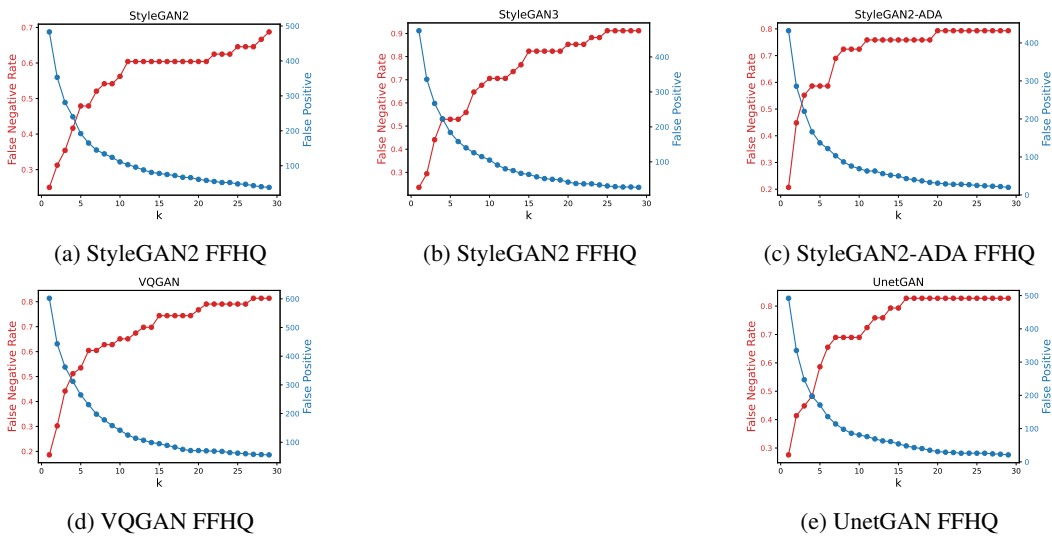

| (a) StyleGAN2 FFHQ | (b) StyleGAN2 FFHQ | (c) StyleGAN2-ADA FFHQ |
| --- | --- | --- |

| (d) VQGAN FFHQ | (e) UnetGAN FFHQ |
| --- | --- |

Figure 22: False Negative Rate (FNR) and False Positive (FP) graphs for the various models trained on FFHQ dataset. FNR refers to the ratio of the number of samples that are not excluded by k-NN among the true out-of-distribution samples, i.e., artifacts. FP refers to the number of samples that are in-distrubtion samples but excluded by k-NN. For the true out-of-distribution samples, we manually labeled 1000 generations into normals and artifacts.

**Choice of** $k$  We show how $k$ for nearest neighbors affects the proposed metric in terms of the Spearman rank correlation. Figure 21 shows the rank correlations among $1 \leq k \leq 10$. The left heatmap shows the rank correlation matrix for the samples included in $M_1$ where $M_i$ indicates **manifold**$_i(\mathbf{\Phi_r})$, i.e., the real manifold built with $i$-NN spheres. The second plot in Figure 21 shows the average rank correlation between $k$ and $1 \leq k' \leq 10$. Each color indicates the target generations in $M_i$. Please note that $M_i \subset M_j$ for $i \leq j$ and the rank correlation changes dramatically for the cases in $M_j \setminus M_i$. For example, for $M_3$, the mean rank correlations have high values $\geq 0.8$ for all $k \geq 3$. While there exists slight drop in the mean rank correlation as $i$ increases, due to the change in the out-of-manifold (OOM) samples, we can see that the mean rank correlation is consistent for $k \geq i$. In other words, the rank of the proposed metric does not change much for different choices of $k$, and we can say the proposed metric is robust to the choice of $k$. However, if we decrease $k$, the number of fake samples out of the real manifold increases as we can see in the graph on the right in Figure 21. Considering that $k \geq 3$ keeps OOM samples under 30%, we set $k = 3$ for the rest of this paper. Further analyses of false-positive versus false-negative-rate in Figure 22 support the choice of $k = 3$.

# H   RESULTS FOR NON-GAN-BASED MODELS

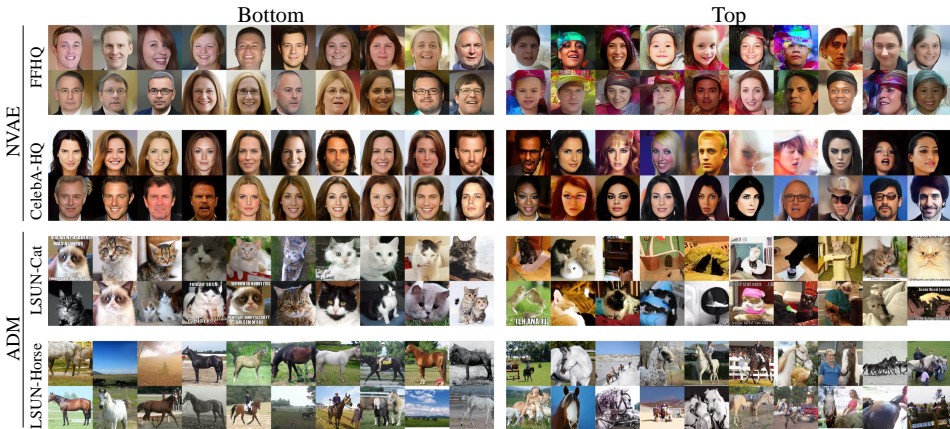

Figure 23: The samples with the lowest rarity scores and the highest rarity scores from NVAEs (Nouveau VAEs) trained on FFHQ and CelebA-HQ, and ADMs (Ablated Diffusion Models) trained on LSUN-Cat and LSUN-Horse, respectively.

# I   USER STUDY

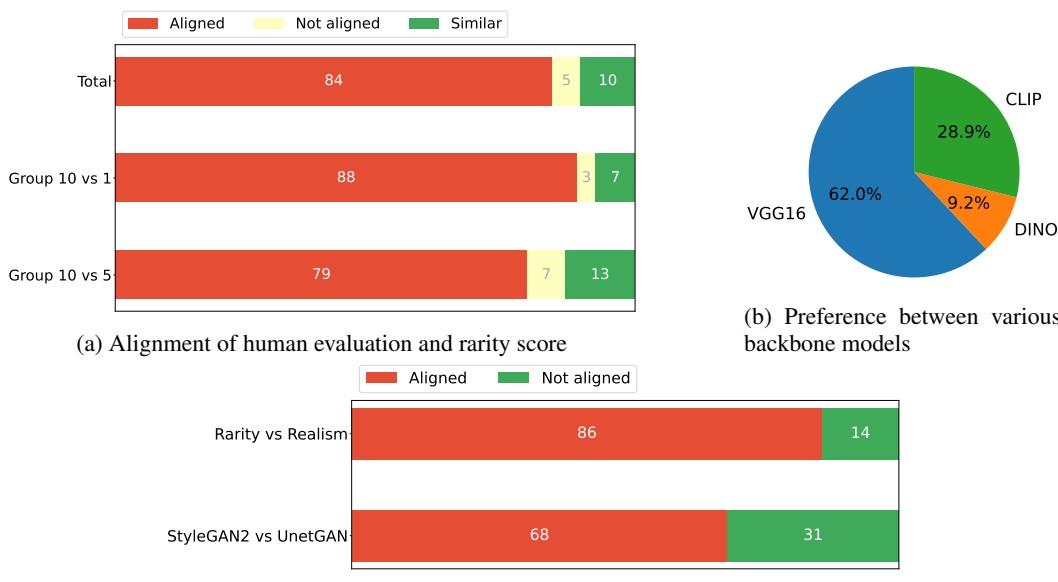

(a) Alignment of human evaluation and rarity score

(b) Preference between various backbone models

(c) Comparisons between rarity score versus realism score and between Style-GAN2 versus UnetGAN trained on FFHQ dataset. For rarity score versus realism score, 'Aligned' represents that rarity is preferred compared to realism score. For StyleGAN2 versus UnetGAN, 'Aligned' represents that Style-GAN2 is preferred compared to UnetGAN.

Figure 24: The summary of the user study.

We have conducted user study whether our proposed rarity score aligns with the human perception of commonness and which backbone model is preferred. 61 participants answered our survey. In this survey, we target on StyleGAN-FFHQ model.

### I.1 HUMAN EVALUATION FOR THE COMMMONNESS

To investigate whether the proposed rarity score aligns with the human perception of commonness/uncommonness, we asked the participants to choose more typical and common images between two sets of images with different levels of rarity scores. Each set contains 9 images. The participants can answer either "A", "B", or "Similar". We divided the generated samples into 10 groups according to its rarity score. Group 1 has the lowest rarity scores and Group 10 has the highest rarity scores. Each participants are asked to answer 10 sets of choices, 5 sets comparing Group 10 and Group 1 and 5 sets comparing Group 10 and Group 5. The order of sets are shuffled and the location of Group 10 (left or right) is also randomly shuffled.

The results are summarized in Figure 24a. In total, 84% of answers were aligned with the rarity score, i.e., human perception of commonness matches with the low rarity scores. For more challenging comparison between Group 10 and Group 5, the alignment is lesser than that of Group 10 and Group 1 and the portion of answer "Similar" increases. This suggests that the proposed metric can distinguish the degree of difference in commonness.

### I.2 PREFERENCE ON BACKBONE

We further asked to the participants which backbone is preferred. For this survey, each participant is given 3 sets of images with different backbones; VGG16, DINO, and CLIP. Each set contains 9 images and the order of backbone models is shuffled each time. Each participant answers 5 times for this type of questions. As a result, VGG16 is the most preferred with 62% of times. CLIP was the second most preferred backbone, but the gap between VGG16 and CLIP was significant.

### I.3 COMPARISON WITH NEGATIVE REALISM SCORE

We additionally conduct a user study on which metric is preferred for measuring uncommonness between rarity score and realism score. We give 10 sets of images, where each set contains top 9 and bottom 9 images for each of the rarity score and realism score. The location (left or right) between rarity score and realism score is randomly selected for each set. For a fairer comparison, we excluded out-of-manifold samples from the samples of the top/bottom realism scores. We ask each participant that which set of images represents rare/common images best. 30 participants answered this survey. The result is summarized in Figure 24c. In total, 84% of answers choose the image set from rarity score compared to the image set of realism score. This supports that rarity score is better suitable for distinguishing and measuring uncommonness realism score, while realism score is useful for measuring fidelity of individual generation.

### I.4 ALIGNMENT WITH RS-p

We further ask participants which model is preferred in terms of rarity between GAN models trained on FFHQ. We choose two GAN models, StyleGAN2 and UnetGAN for the comparison, which have the highest RS-p score and the lowest RS-p score, respectively. We show 10 sets of images, where each set contains top 9 images with the rarity score for each of the StyleGAN2 and UnetGAN. The result is summarized in Figure 24c. 'Aligned' represents that StyleGAN2 is chosen to have rarer images, which is aligned with RS-p score as StyleGAN2 has higher RS-p score compared to UnetGAN. As a result, 68% of answers were aligned with RS-p score. While it is difficult to clearly specify the order of rarity from the generations in Figure 7, the result of user study suggests that RS-p score can guide to choose a model with more rare generations.

## J    NOISE ROBUSTNESS OF RARITY SCORE

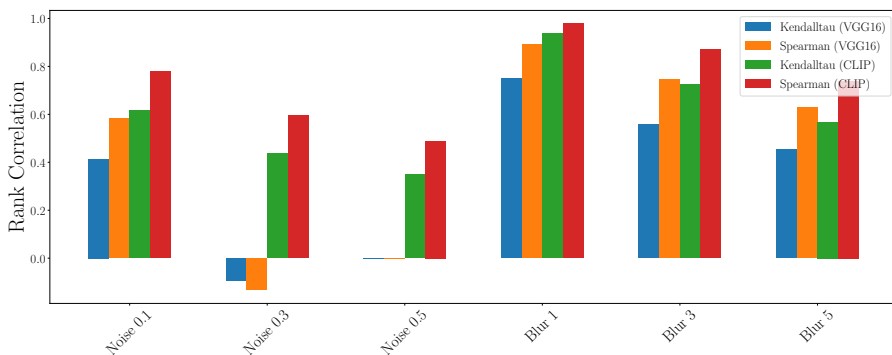

Figure 25: Kendalltau and Spearman's rank correlations between the rarity scores of corrupted images and clean images, according to the noise levels. Noise 0.1, 0.3, 0.5 stand for the standard deviations of Gaussian noise, respectively, and Blur 1, 3, 5 stand for the radii of Gaussian blur, respectively. For VGG16, it is vulnerable to the Gaussian noise, as the rank correlation rapidly decreases and becomes 0 at standard deviation 0.5. This can be improved when we use a backbone more robust to the noise. As an example, CLIP shows better rank correlations compared to VGG16 for every noise level. Both backbones are more robust to the Gaussian blur compared to the Gaussian noise.

## K    RS-P OF FFHQ MODELS WITH VARIOUS BACKBONES

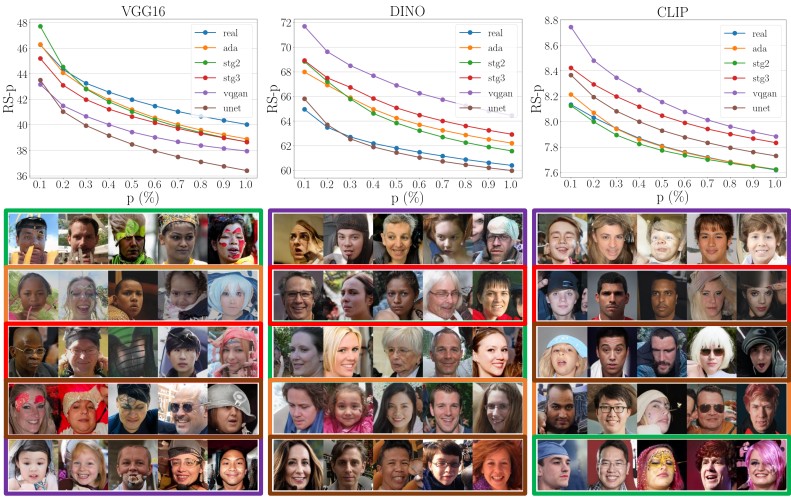

Figure 26: RS-p of FFHQ models and samples with the top 5 rarity scores. The set of samples for each model is marked with color of the bounding box same as the legend color.

We compute RS-p scores of various FFHQ models by adapting the backbone to VGG16, DINO and CLIP. Figure 26 shows the results. As different backbones have different viewpoints of rarity, the order of RS-p also depends on the backbone. For VGG16, StyleGAN2 shows the best RS-p score for its colorful decorations in the generations. On the other hand, for DINO, VQGAN shows the highest RS-p score which have some artifacts on the corner of the image. This is because for DINO, the image with objects other than a human face is regarded as rare. For CLIP, VQGAN again shows the best RS-p score. However, for this time, boys with curly hair have the highest scores. It turns out that, a real image of a magazine cover of a teenage star has the highest nearest neighbor distance and the generations similar to this image get high rarity scores for CLIP backbone.

## L  APPLICATION ON A TEXT-TO-IMAGE MODEL

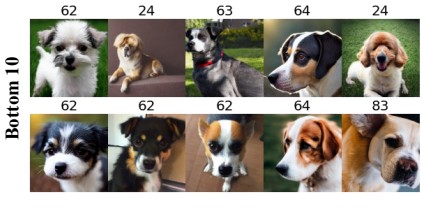

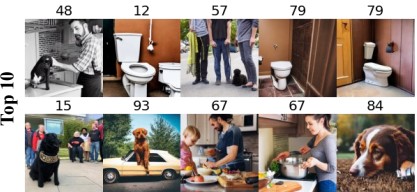

| No. | Prompt (sorted by the occurrence in bottom 10) |
|---|---|
| 62 | the small dog has a tiny black nose. |
| 24 | an image of a dog sitting on a sitting |
| 64 | a dog looking away from the camera in the wind. |
| 63 | a grey dog with a black collar sits outside in the sun. |
| 83 | a close up of a dog near a bowl |

| No. | Prompt (sorted by the occurrence in top 10) |
|---|---|
| 79 | a brown dog standing next to a toilet in a bathroom. |
| 67 | an adult is cooking at the kitchen counter near a baby and a dog. |
| 48 | a man in the kitchen standing with his dog. |
| 12 | a dog is standing next to a toilet playing with tissue |
| 57 | three people stand on the street talking next to a dog. |

Figure 27: The bottom 10 and top 10 generations from Latent Diffusion Model (LDM) and the prompts from MS COCO dataset. The number above each image represents the corresponding prompt number to generate the image.

We compute the rarity scores on a LDM (Latent Diffusion Model), a recently released text-to-image model. For the experiments, we use stable-diffusion-v1-3 from the official git repository. To show an example usage of the proposed metric to a text-to-image models, we sample 2000 real images which contain "dog" in its caption, from LAION-400M dataset, and use 100 prompts containing "dog" from MS COCO 2017 validation dataset for generating fake samples. For each prompt, 10 images are generated from the model. We compute the rarity scores and bottom 10 and top 10 images are shown in Figure 27. We can see that the closed-up dog faces have low rarity scores, which can be regarded as typical dog images. On the other hand, dog images with various backgrounds or environments have high rarity scores. Prompts on the right side of Figure 27 can give some guides the users of text-to-image models, for example, which prompts to use for generating more unique and fun images.

## M  DENSITIES OF COMMON AND RARE SAMPLES

Table 1: KDE restuls for the real samples. 'Low NND' (nearest neighbor distance) represents for the samples with the bottom 1% NND and 'High NND' represents for the samples with the top 1% NND.

| Dataset | Low NND | High NND |
|---|---|---|
| FFHQ | 1585.43 (1166.08) | -14842.84 (8926.26) |
| AFHQ-Wild | 1129.75 (1801.95) | -13723.13 (6822.99) |
| MetFaces | 959.08 (1469.12) | -16448.67 (5709.90) |

We apply the kernel density estimation (KDE) on the real features, to numerically show that the common samples with low NND in Figure 1 have high density in the feature space and the rare samples with high NND have low density in the feature space. We use Gaussian kernel with the bandwidth 0.2. For low NND and high NND, we use 1% of the total real samples, respectively. Table 1 summarizes the mean of log densities for each set of samples. In all datasets, we can see that the samples with low NND, which look typical and common, have significantly higher density compared to the samples with high NND, which have strong characteristics as shown in Figure 1.

# N    ADDITIONAL DISCUSSIONS ON RS-P SCORE

## N.1    CORRELATION BETWEEN RS-P AND OOM RATIO

Table 2: Correlation coefficient between RS-p and OOM Ratio

|  | VGG16 | DINO | CLIP |
|---|---|---|---|
| Correlation Coefficient | -0.23 (0.14) | 0.61 (0.05) | 0.90 (0.03) |

We computed the correlation coefficients between the RS-p scores and OOM ratios of five FFHQ-GAN models. For each p in [0.1, 0.2, 0.3, 0.4, 0.5, 0.6, 0.7, 0.8, 0.9, 1.0], we computed a correlation coefficient from five pairs of RS-p and OOM ratio for each $p$. Please note that the OOM ratio does not change over $p$, given a model. For simplicity, we put mean and standard deviation (inside the parentheses) over $p$ in Table 2. For VGG16, which is the most preferred backbone from the user study, it shows a weak negative correlation. Further, we can see the positive/negative correlation between OOM ratio and RS-p varies over backbones. Thus, it is hard to say that OOM ratio affects RS-p in a consistent manner.

## N.2    CORRELATION BETWEEN RS-P AND FID

Table 3: Correlation coefficient between RS-p and FID Ratio

|  | VGG16 | DINO | CLIP |
|---|---|---|---|
| Correlation Coefficient | -0.82 (0.07) | 0.21 (0.06) | 0.73 (0.05) |

We computed the correlation coefficient between RS-p and FID scores for FFHQ-GAN models in Table 3. The correlation coefficients are averaged over $p$ as in Table 2. For VGG16, we can see a strong negative correlation between RS-p and FID scores. As a lower FID score represents a higher fidelity, the model with high fidelity has high rarity for VGG16. This suggests that rarity and fidelity are not necessarily negatively correlated. However, we cannot say that rarity and fidelity always have positive correlation. For example, RS-p score and FID score show a weak positive correlation for DINO, and a remarkably positive correlation for CLIP, which inversely represents negative correlation between rarity and fidelity. We conjecture that it is caused by the different perspective of rarity depending on used backbones. The feature spaces represented by the backbones are different from each other due to the difference of their learning methods and model architectures (supervised vs. self-supervised or CNN vs. Transformers), and it might make the rarity of each backbone different. As a result, negative/positive correlation may occur depending on the backbone used for feature extraction, and we leave more in-depth analysis as future research.

