# OpenReview forum: "Rarity Score : A New Metric to Evaluate the Uncommonness of Synthesized Images"
_ICLR.cc/2023/Conference — ICLR 2023 notable top 25%_

### Official Review · Reviewer_tiDM · 2022-10-20

**Confidence:** 4
**Correctness:** 3
**Technical Novelty And Significance:** 3
**Empirical Novelty And Significance:** 3
**Recommendation:** 8

**Clarity, Quality, Novelty And Reproducibility:**

**Quality**

I think the quality of experiments can be improved.

**Major: Comparisons between negative realism score**

From Fig 5), one can see that the realism score is inversely correlated with the RS-1 metric, suggesting that the negative realism score may also be used as a “rarity” metric. It would be nice if authors could conduct user studies that show that the “rarity score” is a better measure of rarity as compared to the “negative realism score”.

As a suggestion, the authors can repeat the “user study” done in Section 4.1, but with the “negative realism score” as a baseline. Alternatively, the authors could also take the top x% rare images as given by both the “rarity score” and “negative realism score” and conduct a 2AFC test.

**Major: Handling of Out of manifold**

If for a given sample, a nearest neighborhood manifold cannot be found, then the sample is given an undefined score. How is this handled in the computation of RS-1 score, are the out-of-manifold samples ignored in the cardinality?
Does it make sense to clip it to a lower bound that is defined? (for example, 0.0). In that case, one could just use the mean rarity score of the generated samples. In any case, more details on this are required.

**Major: User studies across models**

The authors perform user studies for one specific GAN, StyleGAN-FFHQ. I think it is also necessary to show that for a given pair of GAN models (A, B), if rarity(A) >rarity(B), the human preference is in accordance with the proposed rarity score.

**Major**

* How does sample quality across different generative models correlate with the rarity metric? Do the authors observe emprically that GAN models with better fidelity also get better rarity scores? It would be nice to show quantitative experiments comparing the relationship between precision/recall, FID and rarity scores for diffferent GAN models.
* "Group 10 (the highest group) and Group 5 (middle group), the alignment is lesser than that of Group 10 and Group 1 (the lowest group) and the portion of answer". These terms are introduced in the main section without any proper defintions. Given that this a paper focusing on a new metrics, I would expect the user studies to be in the main section.
* In Section 4.4, the authors qualitatively show that on using different feature extractors, different images can be considered rare. Quantitatively, for example in Figure 7, does the ranking of GANS also change depending on the feature extractor?
* In Figure 3, 30% real samples are assigned to be out-of-manifold seems pretty high. Can the authors rationalize why this is a good choice? Which dataset was this done on?
* In Figure 5, RS-1 score increases monotonically with the truncation parameter. At which truncation parameter on the x-axis does the correlation breaks?

**Not necessary but nice to have**

* The authors can consider moving the choice of k to the experimental section
* It would be nice to keep the y-axis scale fixed across all three subplots in Figure 5b)
* It would be nice to have some experiments on samples from diffusion models.

**Clarity**

* Page 2, paragraph 2: and helps generative models synthesize rare samples without significantly losing fidelity. I think this is an oversell. The metric is used just to evaluate the samples.
* on top of the various feature spaces with different points of view. What does “feature space with different points of view mean?”. Please modify this sentence.
* LPIPS(Zhang et al, 2018) can be done both instance wise and sample-wise by just aggregating.
* Why is theta is introduced in RS-p, since Phi_g itself refers to samples from that of a generative model in Eq 9.

**Reproducibility**
Code has been promised but it would be nice if authors can provide code snippets or pseudocode.

**Typos**

I spotted some typos. It did not affect my rating but I urge the authors to fix these:

* the effect of the radius is depressed by the effect of the distance between the real and the fake sample. -> Maybe depressed is not the right word here?
* it will be helpful to use rarity score along with the fidelity metric, for selecting images or models in a broader aspect -> I think you can remove broader aspect here.
* Within and outside because the training dataset contains a lot of noisy images for even human to accept. -> i think you can remove even human to accept.
* But also it is -> but also because it is
* alleviates the open challenge problem -> alleviates the open problem
* Generation model performance -> generative model performance
* we can think the sample is extraordinary -> the sample is rare
* is involved in -> is within
* allows us to maintain  -> ensures
* Pg 4, towards the end, real manifold manifoldk(Φr) -> mainfoldk should be bolded
* 30k of real images to approximate the real manifold and calculate the rarity of 10k fake images ->  no of
* answered to -> answered
* generates images more conservatively ->

**Strength And Weaknesses:**

**Strengths**
* While current work on generative model evaluation focuses on sample fidelity and diversity, this paper explores an interesting aspect that is the ability of a model to focus on modeling “niche” images of a dataset.
* The paper is easy to read, is self-contained in the sense that it provides all the required background for the related metrics. The qualitative experiments are extensive and interesting.

**Weaknesses**

My main comment is that the quantitative experiments showcasing the benefits of the “rarity score” can be more rigorous. See below for detailed feedback.


**Summary Of The Paper:**

The paper proposes rarity score, a new evaluation metric for generative models, in order to measure the “rarity” or “uniqueness” of images. The metric builds on prior work such as precision/recall, coverage/density and realism scores that use nearest neighbor manifolds in pretrained VGG-like space to evaluate generative models. The rarity metric of a given image is the radius of the smallest nearest-neighbor manifold that contains the image, provided there exists such a neighborhood. The paper further proposes an extension of rarity score called RS-p to evaluate the rarity of a given generative model.

Quantitative Experiments show that StyleGan2 scores the highest rarity amongst the considered generative models. The authors also perform a number of experiments to qualitatively show rare images. On training datasets, they show that diverse and uncurated datasets like Celeb-A HQ and FFHQ, have a higher proportion of rare images. Amongst generated samples, rare images consist of colorful faces and oil paintings. The rare images obtained also are dependent on the type of feature extractor considered (CLIP, VGG and DINO)


**Summary Of The Review:**

Overall, I think this line of work is promising and interesting. I have some concerns with the experiments and details of the paper. I will consider changing if the authors address the concerns and update the draft.

---

> ### Author Response · Authors · 2022-11-11
> **Author's Response to Reviewer tiDM**
>
> We thank the reviewer for the constructive comments. Below are our detailed response on the comments. If  any concerns remain, please let us know.
>
> ### [Negative Realism Score]
> 4-1. Thank you for the interesting suggestion. We are conducting a user study on the comparison between realism and rarity scores. We will report the results by 11/16.
>
> ### [OOM]
> 4-2. We discard the out-of-manifold samples from the cardinality when computing RS-p.
> For the out-of-manifold samples, assigning a value to out-of-manifold is difficult as both ends of the rarity values [0,inf] have its own meaning as 'common' and 'rare', respectively.
>
> ### [StyleGAN2 vs UnetGAN]
> 4-3. We are conducting a user study on the comparison between StyleGAN2 and UnetGAN. We will report the results by 11/16.
>
> ### [Fidelity vs Rarity]
> 4-4. As most of the recent GAN models show high fidelity, we compare rarity score with FID or realism score changing truncation parameter of StyleGAN2 instead, since truncation shows more dramatic and consistent changes in terms of fidelity (and diversity). As already presented in Figure 5 and the third row of Figure 4, the fidelity improves as the truncation parameter $\psi$ decreases, while diversity decreases. Thus, GAN models with better fidelity may not have better rarity scores. This suggests that a rarity metric can play a complementary role of better assessing a model than using a fidelity metric alone.
>
> ### [User Study in Main]
> 4-5. As the number of pages is limited, we moved user study to the appendix. We will try to move it back to the main section when the undergoing user study is finished by 11/16.
>
> ### [RS-p w/ Different Backbones]
> 4-6.  We are conducting the experiments of RS-p with changing the backbone to CLIP and DINO. We will share the results by 11/13.
>
>
> ### [OOM Ratio]
> 4-7.    StyleGAN2 uses the truncation parameter as 0.5 in the paper for the uncurated qualitative results. This truncation parameter 0.5 will make the sample space shrink by at least 1/2. As we use truncation parameter as 1.0 for the experiments, we think the 30% of loss is tolerable for maintaining the quality of results. For the dataset, FFHQ is used in Figure 3.
>
> ### [RS-1 w/ Truncation Param.]
> 4-8. RS-1 keeps increasing monotonically without breaking correlation until $\psi=1.8$. However, the number of out-of-manifold samples also increases (>80% for $\psi=1.8$).
>
> ### [Writing]
> 4-9. Thank you for your suggestions on the revision. We revised the paper with most of your suggestions.

---

> > ### Author Response · Authors · 2022-11-14
> > **Additional Response to Reviewer tiDM**
> >
> > ### [RS-p w/ Different Backbones]
> > We compute RS-p scores of various FFHQ models by adapting the backbone to VGG16, DINO and CLIP. As different backbones have different viewpoints of rarity, the order of RS-p also depends on the backbone. For VGG16, StyleGAN2 shows the best RS-p score for its colorful decorations in the generations. On the other hand, for DINO, VQGAN shows the highest RS-p score which has some artifacts on the corner of the images. This is because for DINO, the image with objects (i.e., a cat) other than a human face is regarded as rare. For CLIP, VQGAN again shows the best RS-p score. However, for this time, boys with curly hair have the highest scores. We found a real image of a magazine cover of a teenage star has the highest nearest neighbor distance and the generations similar to this image get high rarity scores for CLIP backbone. We put the results in Figure 25 in Appendix K.

---

> > > ### Author Response · Authors · 2022-11-17
> > > **Additional Response to Reviewer tiDM**
> > >
> > > ### [Negative Realism Score]
> > > We additionally conducted a user study on which metric is preferred for measuring uncommonness between rarity score and (negative) realism score. We gave 10 sets of images, where each set contains top 9 and bottom 9 images for each of the rarity score and realism score. The location (left or right) between rarity score and realism score is randomly selected for each set. For fair comparison, we excluded out-of-manifold samples from the samples of the top/bottom realism scores. We asked each participant which set of images represents rare/common images best. 30 participants answered this survey. The result is summarized in Figure 23 (c). In total, 84% of answers chose the image set from the rarity score compared to the image set of realism score. This supports that the rarity score is better suitable for distinguishing and measuring uncommonness than realism score, while realism score more contributes to quantifying fidelity of individual generation.
> > >
> > > ### [StyleGAN2 vs UnetGAN]
> > > We further asked participants which model is preferred in terms of rarity between GAN models trained on FFHQ. We choose two GAN models, StyleGAN2 and UnetGAN for the comparison, which have the highest and the lowest RS-p score, respectively. We show 10 sets of images, where each set contains top 9 images with the rarity score for each of the StyleGAN2 and UnetGAN. The result is summarized in Figure 23 (c). 'Aligned' represents that StyleGAN2 is chosen to have rarer images, which is aligned with RS-p score since StyleGAN2 has higher RS-p score compared to UnetGAN. As a result, 68\% of answers were aligned with RS-p score. User study results suggest that comparison of RS-p scores matches human perception of rarity, and thus RS-p score can help to select a model with more rare generations.

---

> > > > ### Comment · Reviewer_tiDM · 2022-11-22
> > > > **Rebuttal response**
> > > >
> > > > Thanks to the authors for the rebuttal! They have run experiments on comparing the rarity score across GANs from different GAN models and a comparison between the negative realism score which are convincing. I have a few more comments:
> > > >
> > > > 1. Major: I have no concerns with the per-image rarity score. For the per-model rarity score, I am still afraid that just filtering out "Out-of-distribution" images, would bias the metric towards models that generate unrealistic samples. Can the authors confirm that this does not happen in practise? One piece of evidence would be to just report the "fraction of out-of-distribution" samples in Figure 7 and show that there is no correlation between the rarity score and fraction of out-of-distribution samples.
> > > >
> > > > 2. Can you also report the FID of different models in FIgure 7? This will also help see if there is a correlation between the fidelity and rarity across different models.
> > > >
> > > > 3. Please update Appendix I.3 from "Comparison with realism score" to "Comparison with negative realism score".

---

> > > > > ### Author Response · Authors · 2022-11-24
> > > > > **Authors' response to the post-rebuttal comments of Reviewer tiDM**
> > > > >
> > > > > First of all, we appreciate your post-rebuttal comments. As further paper revision is not allowed, we report the results of the experiments with tables here. We will update the title of I.3 in our manuscript.
> > > > >
> > > > > ### [RS-p vs OOM Ratio]
> > > > > **Table a. Correlation coefficient between RS-p and OOM Ratio**
> > > > > |  | VGG16 | DINO | CLIP |
> > > > > |:---:|:---:|:---:|:---:|
> > > > > | **Correlation Coefficient** | -0.23 (0.14) | 0.61 (0.05) | 0.90 (0.03) |
> > > > >
> > > > > We computed the correlation coefficients between the RS-p scores and OOM ratios of five FFHQ-GAN models. For each p in [0.1, 0.2, 0.3, 0.4, 0.5, 0.6, 0.7, 0.8, 0.9, 1.0], we computed a correlation coefficient from five pairs of RS-p and OOM ratio for each $p$. Please note that the OOM ratio does not change over $p$, given a model. For simplicity, we put mean and standard deviation (inside the parentheses) over $p$ in **Table a**. For VGG16, which is the most preferred backbone from the user study, it shows a weak negative correlation. Further, we can see the positive/negative correlation between OOM ratio and RS-p varies over backbones. Thus, it is hard to say that OOM ratio affects RS-p in a consistent manner.
> > > > >
> > > > >
> > > > > ### [RS-p vs FID]
> > > > > **Table b. Correlation coefficient between RS-p and FID**
> > > > > |  | VGG16 | DINO | CLIP |
> > > > > |:---:|:---:|:---:|:---:|
> > > > > | **Correlation Coefficient** | -0.82 (0.07) | 0.21 (0.06) | 0.73 (0.05) |
> > > > >
> > > > > We computed the correlation coefficient between RS-p and FID scores for FFHQ-GAN models. The correlation coefficients are averaged over $p$ as in **Table a**. For VGG16, we can see a strong negative correlation between RS-p and FID scores. As a lower FID score represents a higher fidelity, the model with high fidelity has high rarity for VGG16. This suggests that rarity and fidelity are not necessarily negatively correlated. However, we cannot say that rarity and fidelity always have positive correlation. For example, RS-p score and FID score show a weak positive correlation for DINO, and a remarkably positive correlation for CLIP, which inversely represents negative correlation between rarity and fidelity. We conjecture that it is caused by the different perspective of rarity depending on used backbones. The feature spaces represented by the backbones are different from each other due to the difference of their learning methods and model architectures (supervised vs. self-supervised or CNN vs. Transformers), and it might make the rarity of each backbone different.
> > > > > As a result, negative/positive correlation may occur depending on the backbone used for feature extraction, and we leave more in-depth analysis as future research.

---

> > > > > > ### Comment · Reviewer_tiDM · 2022-11-24
> > > > > > **Thanks!**
> > > > > >
> > > > > > Great, thanks! It seems that atleast for the default feature extractors, interestingly higher fidelity corresponds to higher rarity score, which is interesting.
> > > > > >
> > > > > > I hope the authors can add these insights to the paper if get accepted. If not, in the main draft, atleast a paragraph linking to the corresponding subsections in the supplementary material. I updated my rating to 8.

---

> > > > > > > ### Author Response · Authors · 2022-11-24
> > > > > > > **Thank you!**
> > > > > > >
> > > > > > > Thank you for the update! We appreciate your comments and further discussion. It was our pleasure. We will revise the manuscript with your suggestions!

---

### Official Review · Reviewer_vLSP · 2022-10-24

**Confidence:** 3
**Correctness:** 3
**Technical Novelty And Significance:** 3
**Empirical Novelty And Significance:** 3
**Recommendation:** 6

**Clarity, Quality, Novelty And Reproducibility:**

This paper is clearly written with good quality. Reproducibility is good, more can refer to above comments.


**Strength And Weaknesses:**

strength:
a)	The problem studied in this paper is of high value. There are not many works addressing this problem in current literature. A precise diversity metric is of high demand.
b)	The proposed concept is new, and the implementation is not very complex.
c)	The experiments can mostly support the effectiveness of the proposed metrics.
d)	The organization of the paper is well, and the writing is good.

Weakness & Questions
a)	The proposed metrics do not consider the quality of the generated images. The rarity score is trivially high when the generated images are of low quality. The results of CelebA-HQ with Top 8 in Figure 6 can reflect it. Therefore, the proposed metrics may be biased in this situation.
b)	The density of the real images is not considered in the proposed method. Consider there are two real images, i1 and i2, where the feature-space radius r1 of i1 is much larger than r2 of i2, that is to say, i1 is more rare than i2. Assume that a fake image j which has similar semantics to i1 is in both the spheres of i1 and i2. The rarity score of j is equal to r2 as r2 < r1. The fake image j is then considered as a non-rare image by the proposed metric, which is not consistent with the rarity of real image i1. This may hint the author to consider weighing samples by the density of the related real images.
c)	The hypothesis that “ordinary samples would be closer to each other whereas unique and rare samples would be sparsely located in the feature space”, may need some numerical statistics to support.
d)	Are the proposed metrics robust to the shifts in the input images? (noise, blur, resolution, fidelity, etc.)
e)	I think the results of “Middle” rarity score in AFHQ-Wild in Figure 6 are the most diverse.
f)	From the result of Figure 24, FID seems more accurate regarding correlation (despite negative).
g)	What does the user-preferred backbone reflect? VGG is better than CLIP? Why?


**Summary Of The Paper:**

This paper targets on a fundamental problem in assessing the diversity or rarity of images, especially generated images. The authors first propose a novel evaluation metric, called Rarity Score (RS) to measure image-wise uncommonness, and based on RS, they further propose a model-wise rarity metric RS-p(\theta). The authors show the effectiveness of the proposed metrics through various experiments including a user study.

**Summary Of The Review:**

The studied problem in this paper is important for the vision field. The concept of “rarity score” is interesting. The proposed metric may be impactful if it has more large-scale empirical studies to verify it. Although there are some issues in current version, this is an interesting paper.

---

> ### Author Response · Authors · 2022-11-11
> **Author's Response to Reviewer vLSP**
>
> We thank the reviewer for the constructive comments. Below are our detailed response on the comments.
>
> ### [Fidelity vs Rarity]
> 3-1. Although there are minimum constraints (placed in at least one real knn sphere) for the quality of images, the high-rarity generations might have relatively lower fidelities as rarity score focuses more on the uncommonness rather than on the fidelity. However, as fidelity metrics do not consider the uncommonness themselves, it can be said that the rarity score complements the fidelity metric and vice-versa.
> We have shown the relationship between rarity score and other metrics in Figure5 (a), and we will more emphasize this complementary usage.
>
> ### [Distance between Fake and Real]
> 3-2. Thank you for your constructive suggestion. The cases you mention will be problematic when $i1$ and $i2$ have significantly different rarities. Thus, let’s assume $r2 << r1$. Then, for a fake sample close to $i1$, the fake sample is difficult to be located in both $i1$’s sphere and $i2$’s sphere. It can be possible only when $i2$’s knn sphere is likely to be located in $i1$’s knn sphere. It means that the radius of $i1$’s sphere should be reduced to be similar to the radius of $i2$’s sphere according to the definition of k-nearest neighbor, which contradicts with the assumption $r2 << r1$.
>
>  Empirically, from FFHQ dataset, we observe that in the cases where the distance between the fake sample and $i1$ is smaller than that of $i2$, the maximum difference between the distances is 8.37, where the minimum radius is 10.97 and the maximum radius is 56.84. We also observe that the characteristics of $i1$ and $i2$ are similar in these cases.
>
> ### [Hypothesis]
> 3-3. We are working on how to numerically show that the hypothesis in Figure 1 makes sense. We will share the results by 11/16.
>
> ### [Noise Robustness]
> 3-4. We are conducting experiments by applying Gaussian noise and Gaussian blur to FFHQ StyleGAN2 generations. We will share the results by 11/13.
>
> ### [Diversity vs Rarity]
> 3-5. While “Middle” column of AFHQ-Wild dataset shows diverse species of animals, the “Top” column shows animals with more specific characteristics such as white tigers or female lions, which are less frequently appearing compared to the yellow tigers and male lions in the training dataset.
>
> ### [Trend w/ Truncation Param.]
> 3-6. It looks like FID is well matched with the truncation parameter in Figure 24(a) at first glance. However, considering the definition of FID, it has good (low) scores around truncation parameter $\psi=1.0$ because $\psi=1.0$ is the base distribution for FID calculation. FID score increases (not desirable) as $\psi$ gets larger than 1.0 while diversity will keep increasing as we can see in Figure 5(a) (which is an extended version of Figure 24) and Figure 4.
>
>
> ### [Backbone Preference]
> 3-7. In the user study, VGG16 is the most preferred backbone. We can conjecture this as rarity from VGG16 is the closest to the human perception of rarity for the case of FFHQ dataset. However, we think further in-depth analysis is required to explore the relationships between human perception and feature extractor. We leave it as a future direction because it is out of scope of our paper.

---

> > ### Author Response · Authors · 2022-11-14
> > **Additional Response to Reviewer vLSP**
> >
> > ### [Noise Robustness]
> > We compute the Kendalltau and Spearman's rank correlations between the rarity scores of corrupted images and clean images, according to the various noise levels.
> > For VGG16, it cannot be said robust to the Gaussian noise, as the rank correlation rapidly decreases and becomes 0 at standard deviation 0.5. This can be improved when we use a backbone which is more robust to the noise. As an example, CLIP shows better rank correlations compared to VGG16 for every noise level. Both backbones are more robust to the Gaussian blur compared to the Gaussian noise. We included the results in Figure 24 in Appendix J.

---

> > > ### Author Response · Authors · 2022-11-17
> > > **Additional Response to Reviewer vLSP**
> > >
> > > ### [Hypothesis]
> > > We applied the kernel density estimation (KDE) on the real features, to numerically show that 1) the common samples with low NND in Figure 1 have high density in the feature space and 2) the rare samples with high NND have low density in the feature space. We employed a Gaussian kernel with the bandwidth 0.2 for KDE. For low NND and high NND, we used 1\% of the total real samples, respectively. Table 1 in Appendix M summarizes the mean of log densities for each set of samples. In all datasets, we can see that the samples with low NND, which look typical and common, have significantly higher density compared to the samples with high NND, which are consistent with the characteristics as shown in Figure 1.

---

> ### Comment · Reviewer_vLSP · 2022-12-04
> **feedback**
>
> Thanks for your detailed reply. Most of my concerns are addressed. Please include the points (e.g., limitation, noise test and backbone analysis)into the final version.

---

> > ### Author Response · Authors · 2022-12-04
> > **Thank you!**
> >
> > Thank you for the post-rebuttal feedbacks! We will include the addressed issues in our final version!

---

### Official Review · Reviewer_3DoB · 2022-10-25

**Confidence:** 3
**Correctness:** 4
**Technical Novelty And Significance:** 3
**Empirical Novelty And Significance:** 3
**Recommendation:** 8

**Clarity, Quality, Novelty And Reproducibility:**

Basically, the paper is clear and we can easily follow the idea. Also, the proposed metric is reasonable and novel to me. I think we can easily reproduce it.

**Strength And Weaknesses:**

Strength:

S1: This paper considers an important problem in image generation, proposing a metric to evaluate the rarity of synthetic images.

S2: Extensive experiments are conducted, showing that the proposed metric is reasonable. In addition, user studies make it more convincing.

Weaknesses:

The author only considers the images generated by StyleGAN2, some more advanced models should be considered, in particular the models that employ texts to control image generation. Also, rarity is similar to the outlier, so I think the auth should talk about the relationships between them. Plus, can out-of-distribution or novelty detection approaches be used to evaluate instance-wise rarity?

**Summary Of The Paper:**

This paper pays attention to the vital problem in image generation --- the lack of evaluation metrics. Fidelity and diversity are two common metrics to evaluate synthetic images, while this paper considers another aspect, proposing a metric to evaluate the rarity of synthetic images. Specifically, two metrics are considered --- instance-wise rarity and model-wise rarity. Also, extensive experiments are conducted to show that the proposed metrics are reasonable. Moreover, human studies are conducted, which is more convincing.

**Summary Of The Review:**

The paper considers an important problem in image generation, proposing a metric to evaluate the rarity, which is novel to me.

---

> ### Author Response · Authors · 2022-11-11
> **Author's Response to Reviewer 3DoB**
>
> We thank the reviewer for the constructive comments and encouraging score. Below are our detailed response on the comments.
>
> ### [Text-to-Image model]
> 2-1. In addition to the results of StyleGAN variants, we already presented results on non-GAN-based models such as NVAE (Nouveau VAE) and ADM (Ablated Diffusion Model) in Appendix H. Reflecting the suggestion, we will add the text-to-image results such as Latent Diffusion Model, by 11/16.
>
> ### [Rarity vs. Anomaly Detection]
> 2-2. In general, outliers or OOD samples do not guarantee the quality of samples because they often contain artifacts. In contrast, rarity score limits its target to the in-distribution samples and the high rarity sample is located at the low-density region in the distribution.
> Since anomaly detection or novelty detection approaches target the out-of-distribution samples, it cannot guarantee quality of images. Thus, they are difficult to be an alternative of our rarity score as an uncommonness metric. We will clarify this in the revised version.

---

> > ### Author Response · Authors · 2022-11-17
> > **Additional Response to Reviewer 3DoB**
> >
> > ### [Text-to-Image model]
> > We measured the rarity scores on a LDM (Latent Diffusion Model) **[Rombach et al. 2022]**, a recently released popular text-to-image model. For the experiments, we used [stable-diffusion-v1-3 from the official git repository](https://github.com/CompVis/stable-diffusion). To show an example usage of the proposed metric to a text-to-image models, we sampled 2000 real images which contain "dog" in its caption, from LAION-400M dataset, and used 100 prompts containing "dog" from MS COCO 2017 validation dataset for generating fake samples. For each prompt, 10 images are generated from the model. We compute the rarity scores and bottom 10 and top 10 images are shown in Figure 26. We can observe that the closed-up dog faces have low rarity scores, which can be regarded as typical dog images. On the other hand, dog images with various backgrounds or environments have high rarity scores. Prompts on the right side of Figure 26 can give some hints to the users of text-to-image models, for example, which prompts to be effective for generating more unique and fun images.
> >
> > **[Rombach et al. 2022]** High-Resolution Image Synthesis with Latent Diffusion Models. CVPR 2022.

---

### Official Review · Reviewer_8Kcg · 2022-10-25

**Confidence:** 4
**Correctness:** 3
**Technical Novelty And Significance:** 3
**Empirical Novelty And Significance:** 3
**Recommendation:** 6

**Clarity, Quality, Novelty And Reproducibility:**

Very clear. Very carefully written, organized, and discussed. No concerns on novelty and reproducibility.

**Details Of Ethics Concerns:**

It was not clear whether the authors got permission of the IRB of their intuition.

**Strength And Weaknesses:**

[Strength]
+ The proposed metric, rarity score, is a novel concept.
+ The validity of the proposed metric is quantitatively and qualitatively demonstrated.
+ The metric has multiple applications such as rarity specification when generating images or evaluation of the generation models.
+ The source will be shared.

[Weaknesses]
- Although it is already shown how the proposed model is different from FID and Realism scores, I still suspect the rarity score is counter correlated with the realism score. The authors might want to discuss the difference from it with facts.
- Limitation is not discussed well. Are there any specific cases that have high rarity score but seems common to humans and vice versa? In other words, the authors might want to discuss unsuccessful cases of the proposed model.
- Some discussions on the difference from the previous models are already given, but the motivation why a new metrics is still needed is not clear yet. Can the authors show the cases where the previous metrices do not work well but the proposed one does?
- The experiments include humans. It was not clear whether the authors got permission of the IRB of their intuition.


**Summary Of The Paper:**

This paper proposes a new evaluation metric for image generation using generative and diffusion models: rarity score. The model evaluates how rare (uncommon) the generated images are judging from the standard distribution. The instance-wise rarity score is based on k-NN and the model-wise rarity score is based on CDF. The experiments show its novelty as compared to the previous metrics and the scores are shown to align with the human perception.

**Summary Of The Review:**

Please see my comments above.

---

> ### Author Response · Authors · 2022-11-11
> **Author's Response to Reviewer 8Kcg**
>
> We thank the reviewer for the constructive comments. Below are our detailed response on the comments.
>
> ### [Rarity vs. Realism]
> Realism score measures the relative distance, rather than how rare the related real sample is. Hence, realism score is more close to **precision** (or **fidelity**) rather than to **recall** or **diversity**. This is shown in Figure 4. Samples with low realism scores show low fidelity and contain artifacts. On the other hand, samples with high realism scores have high fidelity. However, it is difficult to say the samples mean typical or common ones. Thus, we believe our rarity metric is more suitable for measuring the rarity of synthetic images. We will revise the manuscript to be more clear. Also, we will also conduct a user study comparing the results of realism score and rarity score and report by 11/16.
>
> ### [Limitations]
> Thank you for pointing out this issue. As our method is based on the training dataset, the results also depend on the dataset. For example, for LSUN-Cat dataset which is less curated and not aligned, the sample with high rarity score may be an image with a noisy background rather than a cat with an unusual appearance. This is caused by the various noisy images in the dataset which are considered as rare, individually. In this case, preprocessing the dataset would improve the quality of the result.
>
> ### [Motivation]
> To the best of our knowledge, rarity score is the first instance-wise uncommonness metric. While previous diversity metrics such as recall and coverage can quantify model performance only, our rarity score can quantify the rarity of each image. Also, this can be useful in the commercial applications, such as stable diffusion-based applications like NovelAI, for selecting more creative and novel images which are unable from existing metrics such as FID or realism score. It can be especially useful to evaluate the decrease in diversity in scenarios where we fine-tune stable diffusion models.
>
> ### [IRB]
> Thank you for raising this issue. We did not perform the IRB process for user study. In fact, it is still less established to be approved by the IRB process for user studies on risk-free tasks such as simple image quality evaluation in computer vision and machine learning research yet, even if HCI and Medical research require the IRB permission in many cases. However, we think CV and ML research communities need to discuss the IRB process for user study and make some guidelines.

---

> > ### Author Response · Authors · 2022-11-17
> > **Additional Response to Reviewer 8Kcg**
> >
> > ### [Rarity vs. Realism]
> > We additionally conducted a user study on which metric is preferred for measuring uncommonness between rarity score and (negative) realism score. We gave 10 sets of images, where each set contains top 9 and bottom 9 images for each of the rarity score and realism score. The location (left or right) between rarity score and realism score is randomly selected for each set. For fair comparison, we excluded out-of-manifold samples from the samples of the top/bottom realism scores. We asked each participant which set of images represents rare/common images best. 30 participants answered this survey. The result is summarized in Figure 23 (c). In total, 84\% of answers chose the image set from the rarity score compared to the image set of realism score. This supports that the rarity score is better suitable for distinguishing and measuring uncommonness than realism score, while realism score more contributes to quantifying fidelity of individual generation.

---

> > > ### Comment · Reviewer_8Kcg · 2022-11-23
> > > **Rensponse**
> > >
> > > Dear authors,
> > >
> > > Thank you for your responses. I am basically satisfied with them.

---

> > > > ### Author Response · Authors · 2022-11-24
> > > > **Thank you!**
> > > >
> > > > Thank you for the post-rebuttal comments! We appreciate your valuable comments which point out important issues. We will clarify these issues in our manuscript!

---

### Author Response · Authors · 2022-11-11
**General response**

Dear all reviewers,

We deeply appreciate the reviewers’ valuable and constructive comments.

Reviewers encourage our paper in terms of a **novel** (8Kcg, vLSP) yet **simple** (vLSP) metric that quantifies the **uncommonness of an individual sample** generated from a generative model. The concept of rarity proposed in the paper investigates an **important** aspect of generative models (3DoB, vLSP, tiDM) for the first time, and is of **high demand** with multiple applications (8Kcg, vLSP). Our **extensive experimental results** are convincing (8Kcg, 3DoB, vLSP, tiDM).

We believe that our proposed metric can contribute to the generative model research community including AI for creativity or the ability to generate in low-density regions as well as AI content creation ecosystem including API providers and customers, for selecting more creative generations which is hard from existing per-model evaluation metrics.

We provide responses for the individual questions in the replies to the individual reviewer.

**[Due Dates for Remaining Issues]**
- 11/13
    * Noise robustness experiments
    * RS-p with various backbones

- 11/16
    * User study to compare rarity score with realism score
    * Text-to-image results
    * Numerical statistics of common and rare samples
    * Experiments on GAN pairs

Once again, thank you very much for all your valuable comments and suggestions.

Authors.

---

### Author Response · Authors · 2022-11-17
**Revised Version Uploaded**

Dear all reviewers,

We have just uploaded the revised version of the paper and the additional answers to individual reviewer.

The revised paper includes followings.

- Noise robustness experiments
  - Appendix J
- RS-p with various backbones
  - Appendix K

- User study to compare rarity score with realism score
  - Appendix I.3
- Text-to-image results
  - Appendix L
- Numerical statistics of common and rare samples
  - Appendix M
- User study on FFHQ-GAN models
  - Appendix I.4

If you have any questions or comments, please do not hesitate to share with us.


Sincerely,

Authors

---

### Author Response · Authors · 2022-11-18
**Discussion reminder**

We sincerely thank you for your effort in reviewing our submission. We gently remind the reviewers that we tried our best to address your concerns via our replies and revision of the manuscript. As the discussion period is nearing the end (today), we would be delighted to hear more from you if there are any further concerns.

---

### Decision · Program_Chairs · 2023-01-20

**Decision:**

Accept: notable-top-25%

**Justification For Why Not Higher Score:**

All reviewers agree this is an interesting contribution, and that the claims are well supported by the experiments. The contribution measures the rarity of generated images, and therefore focuses only on assessing the diversity aspect of generative models. Although this is relevant for researchers on image generative models, the contribution may not benefit a broader audience.

**Justification For Why Not Lower Score:**

All reviewers agree this is an interesting contribution, and that the claims are well supported by the experiments.

**Metareview: Summary, Strengths And Weaknesses:**

This paper was reviewed by four knowledgeable referees. The reviewers found the paper well written, easy to follow and self contained (Kcg, 3DoB, vLSP, tiDM); the concept introduced novel (8Kcg, 3DoB, vLSP); and the evaluation sufficiently extensive to support the claims (8Kcg, 3DoB, vLSP). The reviewers raised concerns w.r.t. the rather unclear motivation to introduce a new metric (8Kcg), the experiments too focused on StyleGAN2 (3DoB), the robustness to shift/blurs/etc unclear (vLSP), the quality of some of the experiments (tiDM), and the limitations of the metric (which were not discussed initially). The authors actively engaged in discussion with the reviewers, provided additional supporting experiments and properly addressed all concerns. After the discussion phase, all reviewers agree that the paper presents an interesting contribution, which appears sound and well validated, and which could be interesting to the community. The AC agrees with the reviewers' assessment and therefore recommends to accept.

**Note From Pc:**

if the above contains the word "oral" or "spotlight" please see: "oral" presentation means -> notable-top-5% and "spotlight" means -> notable-top-25%. As stated in our emails, we are disassociating presentation type from AC recommendations